# NSs amyloid formation is associated with the virulence of Rift Valley fever virus in mice

Psylvia Léger [1,2], Eliana Nachman[3], Karsten Richter[4], Carole Tamietti[5], Jana Koch[1,2], Robin Burk [2], Susann Kummer [2], Qilin Xin[6], Megan Stanifer[2,4], Michèle Bouloy[7], Steeve Boulant[2,4], Hans-Georg Kräusslich[2], Xavier Montagutelli [8], Marie Flamand[5], Carmen Nussbaum-Krammer [3] & Pierre-Yves Lozach [1,2,6 ✉]

Amyloid fibrils result from the aggregation of host cell-encoded proteins, many giving rise to specific human illnesses such as Alzheimer's disease. Here we show that the major virulence factor of Rift Valley fever virus, the protein NSs, forms filamentous structures in the brain of mice and affects mortality. NSs assembles into nuclear and cytosolic disulfide bond-dependent fibrillary aggregates in infected cells. NSs structural arrangements exhibit characteristics typical for amyloids, such as an ultrastructure of 12 nm-width fibrils, a strong detergent resistance, and interactions with the amyloid-binding dye Thioflavin-S. The assembly dynamics of viral amyloid-like fibrils can be visualized in real-time. They form spontaneously and grow in an amyloid fashion within 5 hours. Together, our results demonstrate that viruses can encode amyloid-like fibril-forming proteins and have strong implications for future research on amyloid aggregation and toxicity in general.

[1] CellNetworks—Cluster of Excellence and Virology, University Hospital Heidelberg, 69120 Heidelberg, Germany. [2] Center for Integrative Infectious Diseases Research (CIID), Virology, University Hospital Heidelberg, 69120 Heidelberg, Germany. [3] Center for Molecular Biology of Heidelberg University (ZMBH) and German Cancer Research Center (DKFZ), DKFZ-ZMBH Alliance, 69120 Heidelberg, Germany. [4] DKFZ, 69120 Heidelberg, Germany. [5] Structural Virology, Institut Pasteur, 75015 Paris, France. [6] University Lyon, INRAE, EPHE, IVPC, 69007 Lyon, France. [7] Unité de Génétique Moléculaire des Bunyavirus, Institut Pasteur, 75015 Paris, France. [8] Mouse Genetics Laboratory, Institut Pasteur, 75015 Paris, France. ✉email: pierre-yves.lozach@med.uni-heidelberg.de

Protein misfolding diseases (PMDs) refer to a large array of animal diseases associated with localized or systemic tissue deposits of amyloid fibrils[1]. To date, around 50 human proteins have been identified as distinct amyloid fibril-forming proteins, many giving rise to specific amyloid diseases such as Alzheimer's disease, Parkinson's disease, and Creutzfeldt–Jakob disease[1]. Although the formation and accumulation of amyloid deposits is clearly associated with PMDs, the exact molecular and cellular mechanisms contributing to these diseases remain elusive. PMDs generally arise spontaneously, or to a minor extent, are inherited[2]. An exception are prion diseases that can be acquired through infection[2]. With the increase in life expectancy, many PMDs are becoming increasingly prevalent worldwide. Yet none is curable.

Amyloid-type aggregates result from the misfolding and aberrant assembly of amyloidogenic proteins into highly ordered, linear fibrils[1]. In this process, a partially unfolded monomeric precursor adopts an alternative β-sheet-rich conformation, which is distinct from its soluble native state. The primary nucleation involves the initial aggregation of several of such precursors into smaller either globular or fibrillar oligomeric structures. This step is followed by a rapid growth of oligomers into protofibrils that eventually mature into fibrils by a continuous incorporation of monomers[3]. Depending on the protein involved in fiber formation, up to six protofibrils may further twist around each other in the mature amyloid fibril.

Amyloid deposits that are formed in tissues and cells are highly heterogeneous in size and shape but are usually present in the detergent-insoluble fraction after lysis. In ultrastructure studies they are typically composed of many fibrils that often appear nonbranched, straight, or helical, with a diameter in the range of 10 nm and a length of up to several micrometers[1,3]. While amyloidogenic proteins are host-encoded and present in both prokaryotes and eukaryotes, whether viruses encode such protein remains unclear[4]. Though truncated forms of genetically engineered viral proteins were shown to exhibit some amyloid characteristics in vitro[5,6], no virus-encoded amyloid-like fibrils have yet been described in vivo, i.e., in authentic viral infections in cell model systems or animals and without any genetic or chemical modifications.

We focus here on Rift Valley fever virus (RVFV) and its protein NSs, which assembles into large filamentous structures in the nuclei of infected cells[7]. RVFV is an emerging mosquito-borne pathogen causing serious diseases in both humans and domestic animals[8]. Acute hepatitis and delayed-onset encephalitis are the main features of the most severe cases in humans and are often fatal. At the molecular level, RVFV has a tri-segmented, mainly negative-sense RNA genome that replicates in the cytosol. The viral genome encodes only a few proteins, among them the small NSs protein of about 30 kDa[9].

The viral protein NSs is the main factor of RVFV virulence[10]. Mice infected with recombinant RVFV lacking the NSs sequence (RVFV ΔNSs) survive infection despite production of viral progeny and virus propagation[11]. In contrast, mice inoculated with the wild-type (wt) strain develop acute hepatitis, which is not lethal to animals, but is followed by encephalitis leading to death within a few days[11–13]. Several lines of evidence support that NSs contributes to the disease outcome by modulating host cell functions and defense mechanisms. The protein is a strong inhibitor of type I interferon (IFN)-induced antiviral responses[10] and NSs nuclear filaments are implicated in chromosome cohesion and segregation defects[14]. The existence of nuclear NSs filaments in vivo and a direct role of NSs in the RVFV neurotropism and neurotoxicity have, however, not been demonstrated.

In the present study, we reveal that NSs filaments are bundles of thin fibrils reminiscent of amyloid deposits, not only present in the nuclei of infected cells but also, though shorter in length, in the cytosol. To demonstrate that NSs filaments are indeed amyloid-like, we extensively analyze the ultrastructure and assembly dynamics of NSs in cell monolayers and animals as well as NSs reactivity to amyloid-binding dyes and solubility in detergents. Our results provide a detailed picture of the NSs cell biology and establish NSs as an amyloidogenic protein responsible for RVFV-induced neuropathy symptoms leading to animal death.

## Results

**NSs forms nuclear filaments in a wide range of cell lines.** Many cell types have been reported to support productive RVFV infection but only a few have been used to study nuclear NSs filaments. Therefore, to examine NSs filamentous structures in cell culture, we first assessed the capacity of NSs to form nuclear filaments in ten cell lines, from different mammalian species and tissues. To this end, cells were exposed to the wt virus and analyzed by confocal microscopy after immunofluorescence staining against NSs. Regardless of the tissue and species from which they were derived, all cell lines exhibited NSs and the typical nuclear filaments that have been described for mammalian cells as early as 4–6 h post-infection (pi) (Fig. 1a, Supplementary Fig. 1, and Supplementary Table 1). As expected, NSs and nuclear filamentous structures were absent from cells infected with the genetically engineered RVFV ΔNSs. Infection with this virus was confirmed by the presence of the viral nucleoprotein N in the cytosol.

From the results, it was apparent that all tested mammalian cell lines support efficient formation of large nuclear NSs structures, without noticeable differences in assembly dynamics and filament morphology. For our investigation of the ultrastructure and biology of NSs filamentous assemblies in cell monolayers, we thus selected four lines representing human, monkey, and murine host cells (A549, HeLa, Vero, and L-929) and equally used them in our various imaging methods and other approaches.

**Ultrastructure of nuclear NSs assemblies.** To gain insights into the molecular organization of nuclear NSs filaments, we performed super-resolution and ultrastructural analysis. Three-dimensional (3D)-super-resolution stimulated emission depletion (STED) microscopy was first employed to analyze the nuclei of infected Vero cells. With this approach, we observed the typical nuclear NSs filaments in infected cells (Fig. 1b). They seemed to be composed of many linear subunits (Fig. 1c).

To visualize the ultrastructure of NSs filamentous aggregates, we imaged nuclei of HeLa and Vero cells by transmission electron microscopy (TEM) 6 h pi, when filaments began to become visible. With an average maximal diameter of $634 \pm 158$ nm ($n = 14$) and a length larger than 10 μm, the typical nuclear filaments detected by fluorescence microscopy were easily found (Fig. 1d). Our electron micrographs revealed that they are in fact bundles of thin, straight fibrils reminiscent of amyloidogenic fibrils (Fig. 1e). The width of individual nuclear fibrils was $12 \pm 2$ nm ($n = 34$). Though roughly parallel to each other, the fibrils did not show regular packed alignments, and contacts between fibrils were frequent but not structurally enhanced. When infected cells were subjected to immunofluorescence staining against NSs and analyzed by correlative light and electron microscopy (CLEM), a perfect colocalization could be seen between NSs-associated fluorescence and the fiber-bundles (Fig. 1f). Together, these results show that the thin nuclear fibrils are composed of NSs.

Next, infected Vero cells were analyzed by EM or a method combining STED microscopy and the deep learning-based ilastik software[15]. Compared with our EM-based approach, the STED

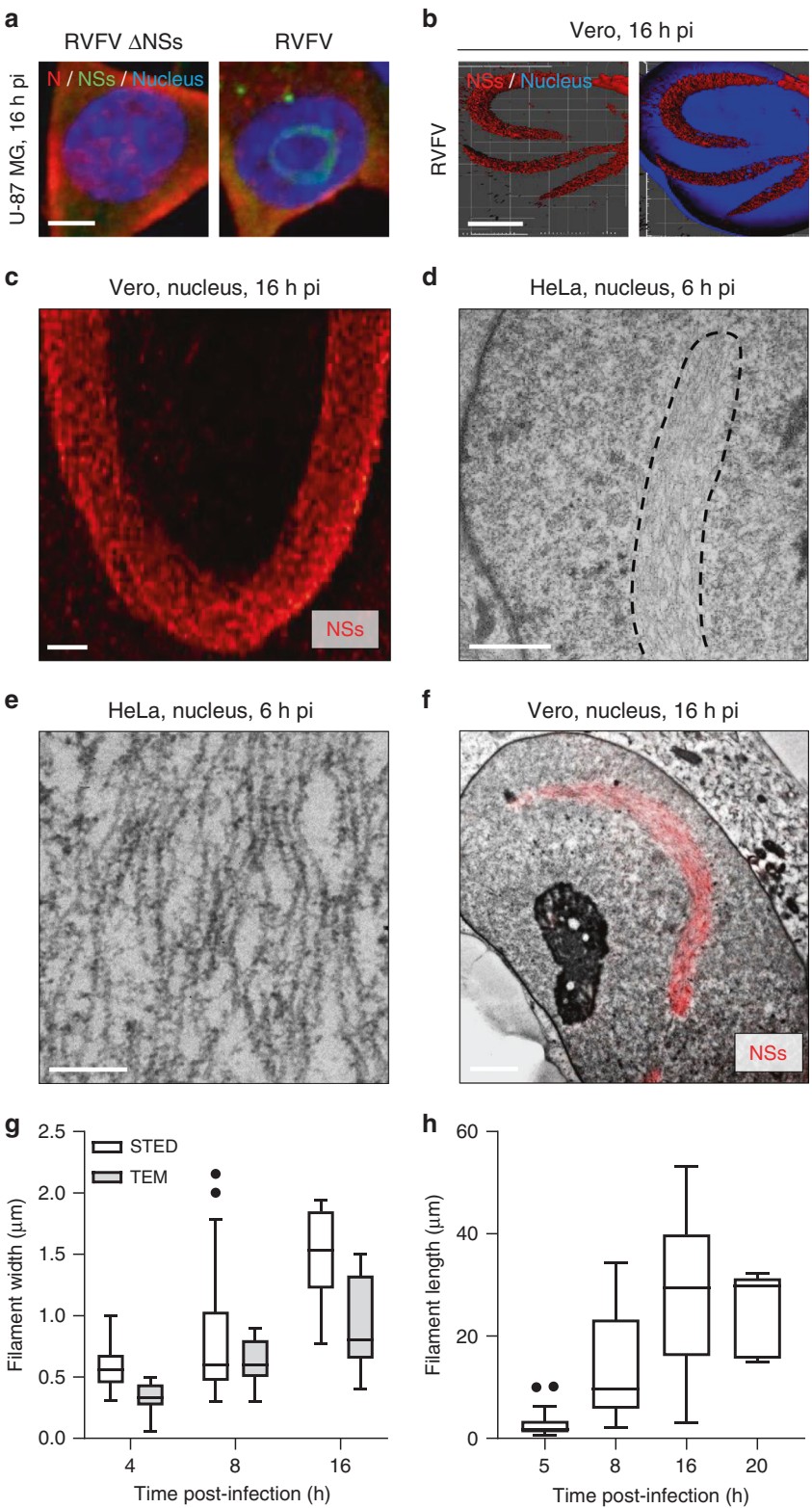

microscopy enabled quantifying the size of each nuclear filament in 3D, as described in Supplementary Fig. 2. At 4–5 h pi, the filaments showed a rather broad distribution in size, with an average width and length of $823 \pm 533$ nm ($n = 14$) and $13.5 \pm 9.9$ μm ($n = 30$), respectively (Fig. 1g, h).

In addition to the highly ordered fibrils, many NSs-positive aggregates with a size of a few tens of nanometers were seen by STED microscopy early after infection of Vero cells, i.e., 5 h pi

(Supplementary Fig. 3A). Using TEM, we frequently observed islands of dense grains with a rosette arrangement and a diameter of $62 \pm 11$ nm ($n = 75$) in HeLa cells 4–6 h pi, exclusively in the nucleus (Supplementary Fig. 3B, arrowheads). Their proximity to the nuclear filaments suggests a connection with the NSs fiber-bundles. None of the nuclear rosettes were found in cells infected with the ΔNSs virus, indicating that all these structural arrangements relate to NSs (Supplementary Fig. 3C). Altogether,

**Fig. 1 Nuclear NSs filaments are bundles of many individual thin fibrils. a** Human U-87 MG cells were exposed to Rift Valley fever virus strain ZH548 (RVFV) or its counterpart devoid of the NSs sequence (RVFV ΔNSs), both at MOI ~5. Infected cells were imaged by confocal microscopy after labeling of nuclei with Hoechst (blue) and immunofluorescence staining of the RVFV proteins NSs (green) and N (red). Images are representative of three independent experiments. Scale bar, 5 μm; pi post-infection. **b** Green monkey Vero cells were infected with RVFV at a MOI of 5 for 16 h. Samples were then stained with Hoechst and antibodies (Abs) against intracellular NSs and imaged by super-resolution stimulated emission depletion (STED) microscopy. 3D-reconstruction of STED Z-stacks was achieved with IMARIS software and shows nuclear NSs filaments in red and the nuclei of infected cells in blue. Experiments were repeated independently three times with similar results. Scale bar, 5 μm. **c** High magnification STED microscopy image of a nuclear NSs filament (red). Results are representative of three independent experiments. Scale bar, 1 μm. **d** Transmission electron microscopy (TEM) of nuclear NSs fibrillary aggregates. Human HeLa cells were infected with RVFV (MOI ~5) and examined in thin sections 6 h pi. The black dashed line outlines one NSs fibrillary aggregate. Images are representative of five independent experiments. Scale bar, 1 μm. **e** High magnification TEM image of nuclear NSs filaments in HeLa cells. Note that filaments are bundles composed of many nonbranched, roughly parallel individual fibrils. Results are representative of five independent experiments. Scale bar, 200 nm. **f** Correlative light and electron microscopy (CLEM) images of nuclear NSs fibrillary aggregates. Vero cells were exposed to RVFV (MOI ~5) and examined in thin sections 16 h pi after immunofluorescence staining against NSs (red). The EM picture shown here was stitched from seven single electron micrographs at 12,500-fold magnification. Experiments were repeated independently three times with similar results. Scale bar, 2 μm. **g** Electron micrographs and STED images of infected Vero cells were analyzed for the width of nuclear NSs filaments. $n = 19, 30,$ and 7 cells and $n = 11, 7,$ and 10 cells were examined by TEM and STED microscopy at 4, 8, and 16 h pi, respectively. Center line, median; box limits, upper and lower quartiles; whiskers, 1.5× interquartile range; points, outliers. **h** Vero cells were exposed to RVFV for up to 20 h and immuno-stained against NSs prior STED microscopy analysis. Images were quantified with the deep learning-based ilastik software. The length of NSs filaments was measured as described in Supplementary Fig. 2. $n = 19, 30, 11,$ and 5 cells were examined at 5, 8, 16, and 20 h pi, respectively. Center line, median; box limits, upper and lower quartiles; whiskers, 1.5× interquartile range; points, outliers.

these results suggest that at least some of the nuclear bodies observed by TEM are built from NSs.

We then examined the ultrastructure of nuclear NSs arrangements at different time points, during the first 20 h of infection. TEM showed that the nuclear filaments were even more prominent 8 h pi, reaching a width of 1 μm (Fig. 1g and Supplementary Fig. 4A). Within these filaments, individual fibrils locally and frequently annealed laterally to denser structures (Supplementary Fig. 4B, arrowheads). At later stages, i.e., 16 h pi, nuclear filaments were larger (about 1.5 μm of diameter) and pervaded substantial parts of infected cell nuclei (Fig. 1g and Supplementary Fig. 4C). At this moment bundles were densely packed (Supplementary Fig. 4D) and most had a length of more than 30 μm, some reaching 40 μm (Fig. 1h). Individual fibrils were difficult to depict, alternating between $8 \pm 2$ nm and $22 \pm 6$ nm ($n = 90$) in width (Supplementary Fig. 4D, arrowhead). This alternation was reminiscent of twists often observed in mature amyloid fibrils that consist of at least two protofilaments, twisted around each other[16].

**The amyloid-binding dye ThS binds to the NSs fiber-bundles.** Our ultrastructural investigation by electron microscopy (EM) and fluorescence microscopy suggested NSs filaments to be amyloid-like. To pursue this possibility, infected cells were subjected to the amyloid-binding dye Thioflavin-S (ThS), which has been broadly used to identify and classify amyloids[17]. This probe displays an enhanced fluorescence at 426 nm upon selective binding to the characteristic β-sheet structure of amyloids. When ThS was applied to cells for confocal microscopy, the dye poorly enters the nucleus. To determine whether ThS binds to the large NSs nuclear filaments, we therefore utilized a Triton X-100-derived buffer to enhance the permeabilization of nuclear envelope and make the nuclear content accessible to the dye. Using this protocol, a strong increase in fluorescence could be visualized in nuclei of cells infected with RVFV by confocal microscopy (Fig. 2a). The fluorescence resulting from ThS binding to nuclear filaments perfectly overlapped with the NSs immunofluorescence signal, proving for the ability of ThS to interact with NSs fibrils. Cells exposed to the ΔNSs virus did not display a fibrillar nuclear ThS signal. The integrity of nucleoplasmic and nucleolar organization was preserved in those cells, and the dye appeared to be trapped in the nucleolus. The fact that no such a signal was observed in cells expressing NSs was because

(i) most nucleoli in the cells had disappeared and (ii) the nucleolar staining was masked due to the strong ThS signal associated with NSs filaments.

**NSs aggregates are resistant to strong detergents.** Many amyloid fibrils are resistant to solubilization by the detergent sodium dodecyl sulfate (SDS). We therefore tested whether NSs shares this property. Nuclear extracts of infected Vero cells were subjected to 2% SDS at increasing temperatures. Analysis of NSs by western blot (WB) revealed multiple bands (Fig. 2b). They included monomers of about 25 kDa, multimers from roughly 50–70 kDa, and an insoluble fraction of high molecular weight (HMW) that remained in the well. The HMW bands were likely to represent the large NSs aggregates and fibrils observed by microscopy. Aggregates dissolved gradually with increasing temperatures until almost completely by heating to 95 °C, albeit the 50–70 kDa multimer fraction remained intact.

To further examine the NSs oligomers of HMW observed by SDS-PAGE, lysates of infected Vero cells were analyzed by semi-denaturing detergent-agarose gel electrophoresis (SDD-AGE), a widely used method to analyze large protein assemblies[18]. Again, samples were incubated with increasing amounts of SDS at increasing temperatures. At 22 °C, solubilization of aggregates was only marginal even after a 2%-SDS treatment and a fraction of NSs aggregates remained clearly visible in the well (Fig. 2c). Increasing the temperature up to 95 °C in the presence of 2% SDS dissolved the largest assemblies, which remained in the wells at lower temperatures, but was not sufficient to completely dissociate the NSs oligomers. From the results, we conclude that NSs fibrils are resistant to strong detergents such as SDS, which is a typical feature of amyloids.

**Disulfide bonds stabilize the large NSs fibrillar assemblies.** The polymerization of the amyloid-like protein MLKL has recently been proposed to depend on disulfide bond formation[19]. NSs contains five cysteine residues. To assess a potential influence of disulfide bonds on NSs assembly formation, lysates of infected Vero cells were treated with 10 mM dithiothreitol (DTT) in the presence or absence of 2% SDS at 22 and 95 °C. Samples were subsequently analyzed by SDD-AGE and immunoblotting using antibodies (Abs) against NSs. While the treatment with SDS again only partially dissolved the HMW assemblies, treatment with

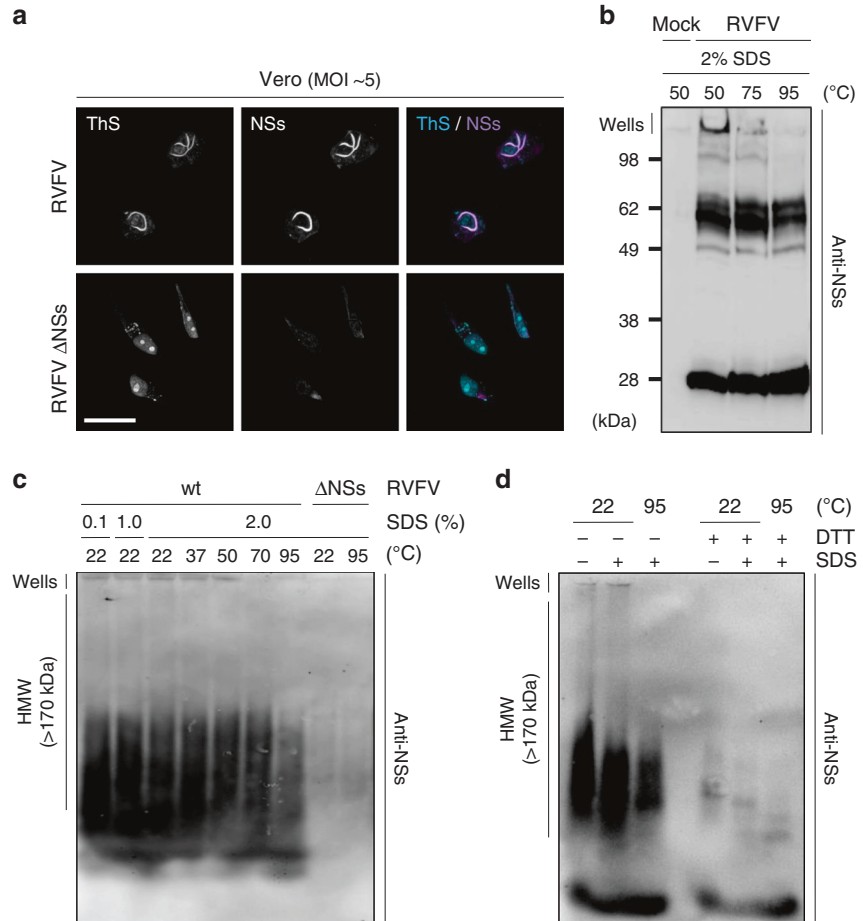

**Fig. 2 NSs aggregates exhibit amyloid properties. a** Vero cells were infected with RVFV and RVFV ΔNSs for 16 h. Infected cells were permeabilized with a Triton X-100-based buffer and then stained with the amyloid-binding dye Thioflavin-S (ThS, blue) and Abs against NSs (purple) before analysis by confocal microscopy. Images are representative of three independent experiments. Scale bar, 20 μm. **b** Vero cells were exposed to RVFV (MOI ~5) and nuclei isolated 16 h pi before being subjected to 2%-SDS treatments and incubated at the indicated temperatures for 15 min. Nuclear extracts were analyzed by 12.5% SDS-PAGE without stacking gel, and under nonreducing conditions followed by WB with Abs against NSs. Results are representative of three independent experiments. **c** Vero cells were infected with RVFV at a MOI of 5 for 16 h and lyzed. Cell lysates were then subjected to increasing amounts of SDS and incubated at the indicated temperature for 5 min. Proteins were analyzed by 1.5% semi-denaturing detergent-agarose gel electrophoresis (SDD-AGE) followed by capillary transfer and immunoblotting against NSs. Experiments were repeated independently three times with similar results. **d** Total lysates of Vero cells infected at a MOI of 5 for 16 h were treated with 10 mM DTT and 2% SDS before being incubated at the indicated temperatures for 5 min. Samples were subsequently analyzed by 1.5% SDD-AGE and WB against NSs. HMW, high molecular weight; wt, wild type. Results are representative of three independent experiments.

DTT led to an almost complete disappearance of the HMW smears (Fig. 2d). Instead, distinct oligomeric bands became visible. When the samples were additionally treated with SDS, their pattern changed again slightly, and an additional prominent presumably monomeric band appeared.

DTT treatments and SDD-AGE analysis suggested that the large NSs fibrillar assemblies are substantially stabilized by disulfide bonds. To pursue this possibility, we replaced each of the cysteine codons by a serine codon in the NSs open reading frame (Supplementary Fig. 5A) and successfully recovered the four corresponding mutant viruses from plasmids, namely RVFV NSs C39S/C40S, C149S, C178S, and C194S (Supplementary Fig. 5B). No major differences were found in the infectivity of the four virus strains (Supplementary Fig. 5C). Only the NSs mutant C39S/C40S displayed a somewhat lower expression compared with the wt protein (Supplementary Fig. 5D).

To evaluate the role of cysteine residues in the assembly of NSs fibrillary structures, cells were exposed to the four RVFV mutants and imaged by confocal microscopy after immunostaining against NSs. In agreement with a recent report[20], no filament but large

spheroid aggregates were found in the nuclei of infected cells by RVFV NSs C39S/C40S (Fig. 3a). Similarly, cells exposed to RVFV NSs C149S did not have large filaments but nuclear globular aggregates, more numerous and smaller than those of the NSs mutant C39S/C40S (Fig. 3b, c). The substitution of cysteine residues at position 178 and 194 had no significant effect on the shape, number, and size of the nuclear filaments (Fig. 3a, d, e). In the cytosol of infected cells, all four NSs mutants formed globular aggregates, similar in shape to those consisting of the wt protein (Fig. 3a, c). An increased number of cytosolic assemblies was however measured in cells expressing the NSs mutants C39S/C40S and C149S (Fig. 3f).

To visualize the ultrastructure of aggregates made of the two mutants, which are unable to form large nuclear filaments, infected cells were examined for NSs by immunogold labeling EM. With this approach, gold beads were seen to decorate the typical large nuclear NSs fiber-bundles (Fig. 3g). The spheroid aggregates of NSs C39S/C40S were easily found in the nucleus as regular assemblies with a diameter of 604 ± 316 nm ($n = 17$) and were essentially built of small clumps (Fig. 3g). Interestingly,

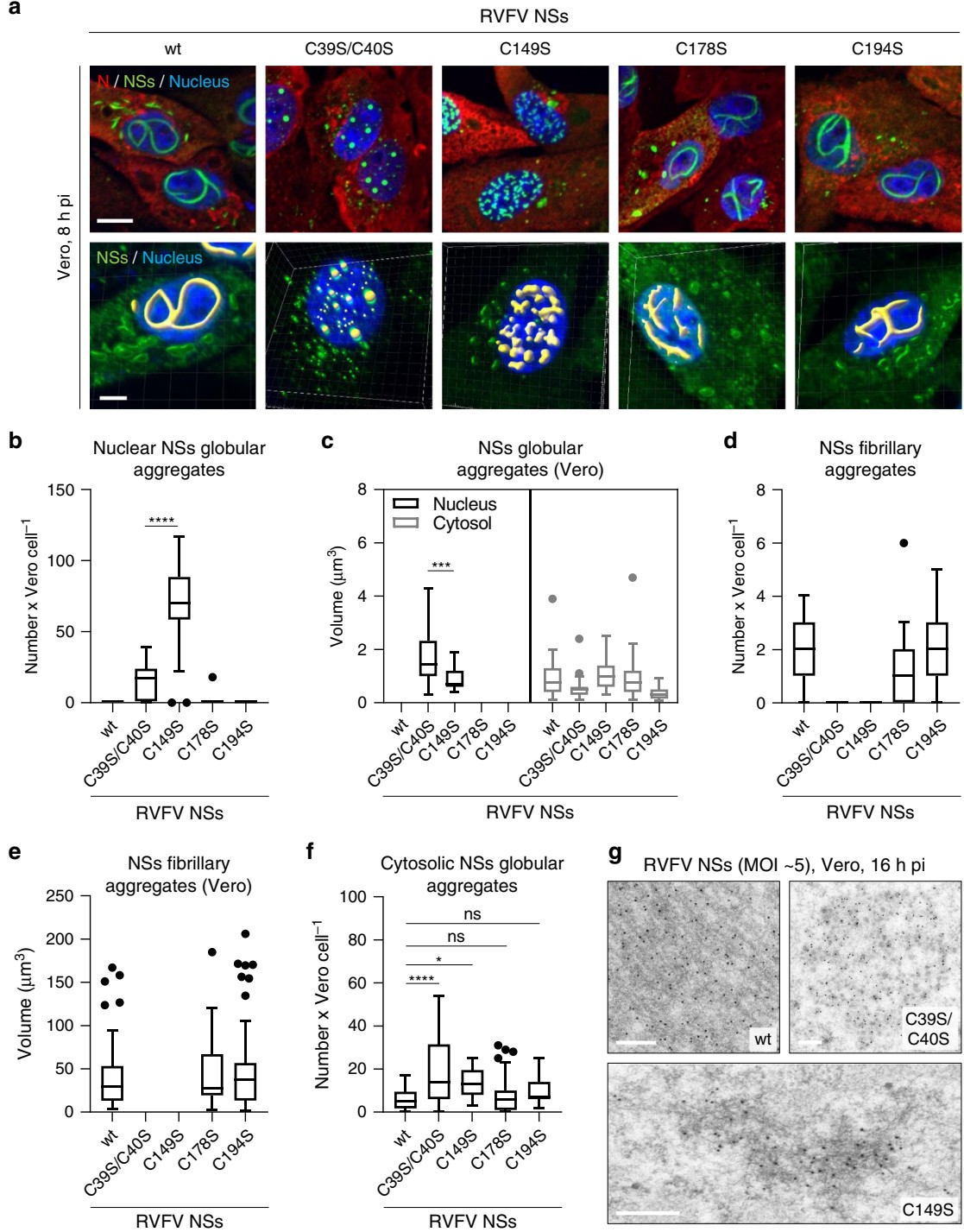

imaging of the globular aggregates formed by the NSs mutant C149S revealed patches of short fibrils with an average length of 4.1 ± 2.2 μm (n = 89) (Fig. 3g). Electron micrographs showing wider fields of view are available in Supplementary Fig. 6. Overall, our results suggest a role of redox mechanisms in the formation of fibrils and the subsequent assembly into large nuclear filaments. Cysteines in the N-terminal region likely drive the initiation of NSs fibrilization, while the one at position 149 the growth of NSs fiber-bundles.

**Nuclear NSs fibrils grow in an amyloid fashion.** Amyloid aggregates have in common to expand by a continuous incorporation of monomers of the amyloidogenic protein rather than

by an increase of the structure number[3]. To test whether NSs also meets this criterion, we assayed the expression of the total NSs by WB in cells exposed to various multiplicities of infection (MOIs) of RVFV with the goal to correlate the level of protein with the number and size of nuclear fiber-bundles. The level of NSs increased with increasing MOIs (Fig. 4a). The production of the protein did not reach a plateau value under our experimental conditions, though a threefold-to-fivefold increase in protein abundance was measured between the lowest and highest MOIs (Fig. 4b and Supplementary Fig. 7A). Immunofluorescence confocal microscopy of infected cells confirmed the positive correlation of NSs expression with MOIs (Fig. 4c). Note that NSs filamentous aggregates increased in thickness at the highest MOI

**Fig. 3 NSs fibrilization relies on specific cysteine residues. a** Vero cells were infected with RVFV and four mutant viruses each coding for NSs with cysteine substitutions, namely C39S/C40S, C149S, C178S, and C194S, for 8 h. Infected cells were analyzed by confocal microscopy after staining of nuclei with Hoechst (blue) and Abs against intracellular N (red) and NSs (green). 3D-modeling of NSs fibrillary and globular aggregates were achieved with the IMARIS software and are shown underneath each respective virus strain. Images are representative of three independent experiments. Scale bars, 5 μm. **b, d, f** Confocal Z-stack obtained in **a** were classified, segmented, and analyzed with the software ilastik as described in Supplementary Fig. 2. The number of NSs aggregates is given per cell as follow: **b** nuclear globular aggregates, **d** nuclear fibrillary aggregates, and **f** cytosolic globular aggregates. $n = 29, 33, 25, 35$, and 31 cells exposed to the wt virus and the mutants C39S/C40S, C149S, C178S, and C194S were respectively examined. Unpaired $t$-test, two-tailed, with Welch's correction [parametric, not equal standard deviations (SDs)] was used in **b** while multiple comparisons, one-way ANOVA tests with Tukey corrections was employed in **f**. *$p = 0.0288$; ****$p < 0.0001$; ns non-significant. Same as in **b, d, f** but showing the volume of NSs globular **c** and fibrillary **e** aggregates. In **c**, $n = 26$ and 23 cells were examined for nuclear NSs globular aggregates (black) made from the NSs mutants C39S/C40S and C149S while $n = 24, 32, 25, 28$, and 30 cells were analyzed for cytosolic globular aggregates (gray) of NSs wt, C39S/C40S, C149S, C178S, and C194S, respectively. In **e**, $n = 51, 50$, and 66 nuclear NSs fibrillary aggregates built from NSs wt, C178S, and C194S were respectively examined. Unpaired $t$-test, two-tailed, with Welch's correction (parametric, not equal SDs) was applied in **c**. ***$p = 0.0002$. **b–f** Center line, median; box limits, upper and lower quartiles; whiskers, 1.5× interquartile range; points, outliers. **g** Vero cells were exposed to RVFV and the two mutant viruses NSs C39S/C40S and C149S (MOI ~5) for 16 h. Electron micrographs show the ultrastructure of wt NSs fiber-bundles (upper left panel) and globular aggregates made of the NSs mutants C39S/C40S (upper right panel) and C149S (lower panel) after immunogold labeling with Abs against NSs (black spots). Experiments were repeated independently thrice with similar results. Scale bars, 500 nm.

(Fig. 4c, white arrowheads), i.e., when the level of NSs was maximal. In these experiments, the viral nucleoprotein N served as an additional readout of infection, as all infected cells abundantly express this protein. Overall, an earlier onset of filament formation correlated with higher NSs expression and MOIs (Supplementary Table 1).

We next analyzed the confocal images obtained after fixation of infected cells and correlated the number and size of NSs structures to the amount of soluble protein determined by WB. Using this approach, we found that the number of nuclear filamentous aggregates did not correlate with the quantity of protein (Fig. 4d and Supplementary Fig. 7B). A maximum number of 3–4 fibrillary aggregates per nucleus was already reached at the lowest NSs level. We also found that nucleoplasmic arrangements became larger as the production of NSs increased (Fig. 4e). Together, these data suggest that a high expression of NSs results in an increased size of the nuclear structures rather than in a higher number.

To analyze the dynamics of aggregation, we monitored NSs in infected cells over time. To this end, cells exposed to the virus were fixed at different time points up to 16 h pi, immuno-stained against NSs, and imaged by confocal microscopy. Nearly all infected cells were positive for NSs and an average of five nuclear filamentous structures were visible as early as 6 h pi in L-929 cells (Fig. 4f). The total number of filamentous aggregates rapidly reached a plateau value, with six fiber-bundles after a further 2 h. Then the filamentous aggregates continued to grow until the cells died (Fig. 4g).

Next, we monitored NSs aggregation in living Vero cells via time-lapse video microscopy. Tagging the N- or C-terminal regions of NSs with large tags, such as fluorescent proteins, was unsuccessful, i.e., the protein expression was compromised. Therefore, we developed an approach based on the small tetracysteine peptide (tc)[21] to tag the N-terminal region of NSs (tc-NSs) (Fig. 5a) and visualize NSs in real-time. The recombinant virus was successfully rescued from cDNAs (Fig. 5b). No significant impact of the peptide on RVFV infectivity was found as the titer was similar to that of the parental virus strain (Fig. 5c). In addition, the tagged protein was as efficient to repress the *IFN-β* gene promoter as the wt protein (Fig. 5d). Therefore, we assume that the tc-NSs variant fulfills the functions of the wt molecule.

The membrane permeable dye ReAsH specifically interacts with the tc peptide and turns fluorescent only following binding to the tag (Fig. 5e)[21]. The high density of tc-NSs molecules in aggregates provided numerous binding sites for the ReAsH dye, offering optimal conditions for the detection of NSs aggregates.

Only a small amount of dye was required to image the formation and motion of nuclear NSs filamentous structures in living cells (Supplementary Movie 1). In addition, the use of a little quantity of ReAsH greatly reduced the background noise and cytotoxicity, usually the main limitations for employing this dye. The typical nuclear filamentous aggregates were visible in infected cells within 4.5 h. Our video confirmed that the NSs fiber-bundles expand in size but not in number (Fig. 5f, g). These results were consistent with our observations in fixed cells (Fig. 4 and Supplementary Fig. 7), indicating that the peptide tag and dye did not introduce any bias in the dynamics of NSs assembly into filaments.

Altogether, the results show that nuclear NSs fibrils grow according to a process consistent with amyloid fibril formation, which develop by recruiting and incorporating monomers into growing fibrils.

**Cytosolic NSs aggregates consist of short fibrils.** Current knowledge of NSs structures focuses on the conspicuous nuclear filaments, the hallmark of RVFV infection. From our results, it was apparent that NSs also aggregates in the cytosol, late after the infection of cells (Fig. 4c, red arrowheads, and Supplementary Movie 1). However, the cytosolic NSs aggregates had a globular aspect and were 10- to 20-fold smaller than those in the nuclei. These stark differences raised the question whether globular aggregates reflect an alternate, nonamyloid structural assembly of NSs in the cytosol of infected cells. To test this possibility, we first defined the assembly dynamics of the NSs globular aggregates in infected HeLa cells. The cytosolic NSs aggregates shared with the nuclear fiber-bundles the property to expand in size, but not in number (Fig. 4h and Supplementary Fig. 7C). A maximum of 3–5 globular aggregates was reached in the cytosol at the lowest NSs level. Similarly, the number of such aggregates did not significantly increase over the period of infection and remained within 5–6 per cell (Fig. 4i).

We then examined the structure of the cytosolic NSs globular aggregates in Vero cells 16 h pi. When infected cells were fixed and subjected to immunofluorescence staining, confocal microscopy images showed the cytosolic expression of the viral nucleoprotein N and the massive nuclear NSs filaments (Fig. 6a). At this late stage of infection, the cytosolic globular aggregates of NSs were easily found and heterogeneous in size. These aggregates were not found in cells infected with RVFV ΔNSs, suggesting that the cytosolic structures consist of NSs as well. When the cytosol was imaged by STED microscopy, aggregates of a few micrometers were also observed as early as 8 h pi (Fig. 6b).

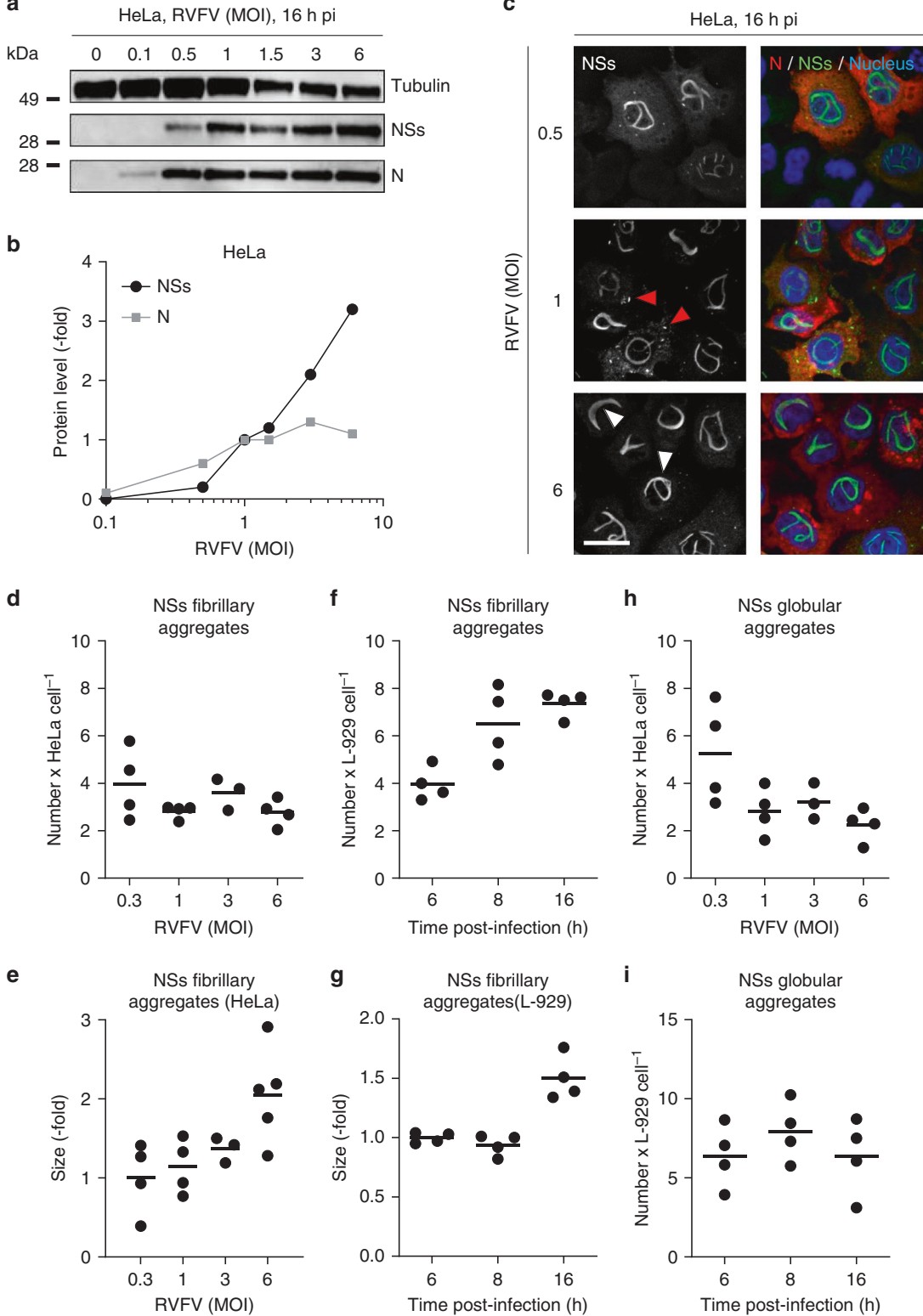

Thereafter, they increased in length and volume until reaching many micrometers in length 12 h later (Figs. 6c, d).

To address the ultrastructure of the NSs globular aggregates, Vero cells were exposed to RVFV for 16 h and imaged by TEM. Our electron micrographs revealed short fibrillar aggregates in the cytosol (Fig. 6e, red arrowheads). They were significantly thinner and shorter compared with those found in the nuclear filaments. With a width of $5 \pm 2$ nm ($n = 23$) and a length of a few hundred

nanometers, these fibrils did not resemble any of the known endogenous regular structures involving actin, tubulin, vimentin, and cytokeratin. Again, such cytosolic aggregates were not found in cells exposed to RVFV ΔNSs. By CLEM, we could observe that the NSs fluorescence signal indeed correlates with the fibrillar aggregates of the cytosol (Fig. 6f). Overall, these results show that although NSs forms aggregates with two distinct morphologies when examined in fluorescence microscopy—filamentous

**Fig. 4 NSs aggregates grow in an amyloid fashion. a** HeLa cells were infected at various MOIs of RVFV for 16 h and assayed for the viral N and NSs proteins by western blot (WB) after 2%-lithium dodecyl sulfate treatment and boiling. **b** Shows the semi-quantification of the N and NSs expression obtained from WB analysis as shown in **a**. N and NSs protein levels are expressed as the fold increase relative to their respective level in cells infected at a MOI of one and normalized to the level of tubulin. Experiments were repeated independently three times with similar results. **c** HeLa cells were exposed to RVFV at the indicated MOIs followed by immunofluorescence staining against N (red) and NSs (green) 16 h later. Nuclei were stained with Hoechst (blue) before confocal microscopy imaging. White and red arrowheads indicate NSs filamentous and globular aggregates, respectively. Images are representative of three independent experiments. Scale bar, 20 μm. **d**, **h** Confocal Z-stack obtained in **c** were classified, segmented, and analyzed with the software ilastik (Supplementary Fig. 2). The number of fibrillary (**d**) and globular (**h**) aggregates is given per positive cell for MOIs ranging from 0.3 to 6. Points represent independent experiments ($n = 4$, with the exception of MOI ~3, $n = 3$). Center line, mean. **e** Scatter dot plot depicting the size of NSs fibrillary aggregates in HeLa cells 16 h pi according to the MOI. The size is expressed as the fold increase relative to the average size in cells infected at a MOI of 0.3. Points represent independent experiments ($n = 4$, with the exception of MOI ~3, $n = 3$, and MOI ~6, $n = 5$). Center line, mean. **f**, **g**, **i** L-929 murine cells were infected with RVFV (MOI ~3) for up to 16 h, immuno-stained against the viral proteins N and NSs. Infected cells were analyzed for the number (**f**, **i**) and size (**g**) of aggregates in NSs-positive cells over 16 h as described in Supplementary Fig. 2. The size in G is expressed as the fold increase relative to the average size measured 6 h pi. Points represent independent experiments ($n = 4$). Center line, mean.

---

aggregates in the nucleus and globular aggregates in the cytosol—both consist of NSs fibrils and resemble amyloid deposits in their ultrastructure.

**NSs alone makes amyloid-like fibrillary aggregates**. We next assessed whether NSs can form amyloid-like aggregates in the absence of any viral infection and additional virus factors. Cell transfection of plasmid DNAs has been shown to be inefficient to express NSs[22]; the protein indeed downregulates mammalian DNA promoters in general[14]. In accordance with these observations, when we transfected Vero cells with the plasmid vector pCI coding for NSs, only 15% of cells expressed the protein (Fig. 7a). However, in all NSs-positive cells, we could observe large nuclear fibrillary assemblies and cytosolic globular-looking aggregates (Fig. 7b), both comparable in size to those assessed in infected cells (Figs. 3c, e and 7c, d). No specific signal could be detected in cells transfected with the empty plasmid, indicating that the nuclear and cytosolic aggregates were made of NSs. Similar results were obtained when other cell lines were transfected (Fig. 7a), with the highest expression of NSs observed in HeLa cells, i.e., about 30% were positive for NSs aggregates.

The amyloidogenic nature of the NSs fiber-bundles obtained by transfection was confirmed using the amyloid-binding dye ThS and confocal microscopy (Fig. 7e). That the ThS fluorescence colocalized with the NSs immunofluorescence signal demonstrated that ThS interacts with NSs aggregates. Together, these results indicated that the NSs protein alone can form amyloid fibril in mammalian cells.

**NSs fibrils suppress IFN responses**. NSs has a central role in RVFV evading IFN responses[10]. NSs is a strong silencer of IFN-β messenger RNA (mRNA)[10], and it mediates the degradation of the protein kinase R (PKR)[23,24], a key player in the IFN response. To determine which of the NSs structural assemblies is required for repressing the IFN-β response, we analyzed the ability of the two NSs mutants C39S/C40S and C149S to suppress the IFN-β mRNA expression in comparison with the wt protein. When A549 cells were infected with RVFV NSs C39S/C40S, which forms clumps but neither fiber-bundles nor short fibrils, IFN-β mRNA expression increased as much as 15-fold over a period of 16 h (Fig. 8a); slightly less than in cells exposed to RVFV ΔNSs. Conversely, NSs C149S, the mutant that assembles into short fibrils but not into fiber-bundles, silenced the IFN-β gene with the same efficiency than the wt protein, i.e., IFN-β mRNA expression remained identical to that in the noninfected control.

The capacity of short NSs fibrils to impair IFN-induced immune defense was confirmed by immunoblot analysis using PKR as a readout. PKR is normally poorly or not at all expressed

in cells infected by RVFV. As expected, Vero cells exposed to the wt virus did not express PKR at 16 h pi, and only marginally earlier (Fig. 8b, c). The short fibrils of NSs C149S degraded PKR as efficiently as the wt viral protein. In contrast, when cells expressed the NSs mutant C39S/C40S, a significant level of PKR was detected, equivalent to that observed in cells infected with RVFV ΔNSs (Fig. 8d). A similar amount of the nucleoprotein N was detected in infected cells regardless of the RVFV strain used, confirming an equal level of infection. Altogether, our data demonstrate that RVFV relies on NSs fibrilization, but not on the emergence of large fiber-bundles, for suppressing IFN responses.

**NSs fibrilization occurs in the brain of infected animals**. NSs is considered as the major virulence factor of RVFV, and neuropathy is one major symptom that characterized RVFV infection in humans and other mammalian hosts[8,10]. BALB/c mice constitute an interesting animal model to investigate RVFV-induced diseases. These animals recapitulate the acute hepatitis, not lethal, and delayed-onset encephalitis, leading to death in severe human cases of Rift Valley fever[12,25]. To determine whether NSs forms nuclear filaments in the brain of infected animals, mice were inoculated intraperitoneally with 100 plaque-forming units (pfu) of the virulent RVFV strain or the natural mutant clone 13 (RVFV ΔNSs C13), which has a large deletion in the NSs sequence and therefore lacks NSs expression[26]. Like the genetically engineered ΔNSs viruses, RVFV clone 13 is avirulent despite being not compromised in infectivity[26]. Infected mice were monitored daily for 10 days. In agreement with previous studies[27], most animals exposed to the wt virus developed signs of illness from day 6 to 8 post-inoculation, while mice inoculated with RVFV ΔNSs C13 did not show any clinical symptoms during this period.

Animals were sacrificed when the first signs of disease appeared. Brains were collected, fixed, and embedded in resin for subsequent thin sectioning and immuno-histochemical staining against NSs and the viral nucleoprotein N. Neither specific N nor NSs staining could be detected in mice inoculated with RVFV ΔNSs C13 (Fig. 9a), indicating that brain cells were not infected. In contrast, strong staining for both viral proteins was observed in the brain of animals infected with the wt virus. Infection appeared to be restricted to specific areas, mainly in the brainstem (Fig. 9b). Together the data indicate that NSs is not only essential for the virulence but also for neurotropism.

To assess the structure and subcellular location of N and NSs in the brainstem of mice challenged with RVFV, thin sections of resin-embedded brains were immuno-stained against the two viral proteins and imaged by confocal microscopy. The RVFV nucleoprotein N displayed a homogenous distribution throughout the cytosol of infected neural cells (Fig. 9c and Supplementary

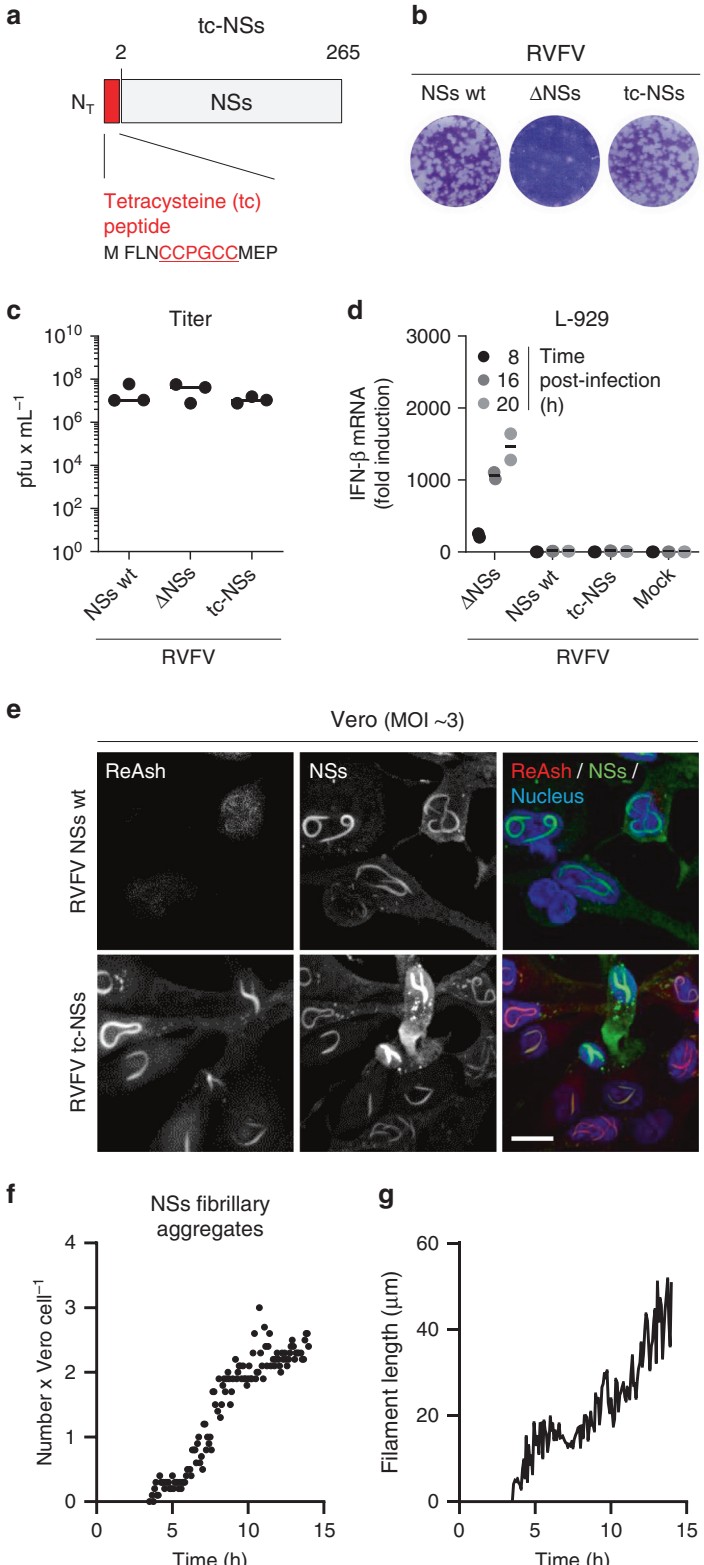

Movie 2) while large NSs filamentous structures were easily visible in the nuclei (Fig. 9d and Supplementary Movie 3).

To confirm that NSs filaments are of amyloidogenic nature in vivo, we stained thin sections of infected brains with the amyloid-binding dye ThS. Consistent with our observations in cell monolayers, staining of the nucleus content was inefficient with ThS. We therefore used a combination of harsh permeabilization treatment with a concentration of ThS as high as 1%. Though this concentration of dyes resulted in an increased background noise over the tissue section, immunofluorescence staining against NSs showed that ThS exclusively associated with NSs-containing filaments (Fig. 9e). The combined observations demonstrate that NSs forms amyloid-like fibrillar aggregates in the brain of infected animals that developed neuropathy symptoms.

**Fig. 5 Recovery and characterization of RVFV encoding tetracysteine (tc)-NSs. a** Schematic depiction of NSs N-terminally tagged with a tc peptide (tc-NSs). **b** Titration of the recombinant RVFV coding for tc-NSs (RVFV tc-NSs) in a monolayer of Vero cells by plaque-forming assay. After 5 days of incubation at 37 °C, plaques were colored with crystal violet. RVFV and its mutant lacking the full sequence coding for NSs (RVFV ΔNSs) were used as controls. wt, wild type. **c** Titer of the genetically engineered RVFV tc-NSs after rescue and five passages in Vero cells. Points represent titers of independent virus productions (*n* = 3). Center line, mean. pfu plaque-forming units. **d** The interferon-β (IFN-β) mRNA levels were quantified by real-time quantitative reverse transcription PCR (qRT-PCR) in L-929 cells infected at MOI ~3 with the indicated viruses for up to 20 h. Points represent replicates (*n* = 2). Results are representative of three independent experiments. Center line, mean. **e** Vero cells were exposed to RVFV and RVFV tc-NSs (MOI ~3) and subjected to immunofluorescence staining against NSs (green) 16 h pi. Tc peptide and nuclei were stained with ReAsH (red) and Hoechst (blue), respectively, prior to imaging by confocal microscopy. Images are representative of three independent experiments. Scale bar, 15 μm. **f, g** RVFV tc-NSs (MOI ~5) was added to Vero cells for 3 h, and after replacing the input virus by phenol red-free medium containing the ReAsH dye, infected cells were imaged in real-time with a wide-field microscope at 37 °C for up to 14 h. Images were taken every 5 min, and each frame was analyzed as described in Supplementary Fig. 2. The number of NSs fibrillary aggregates per cell (*n* = 50) and the average length of NSs filaments (*n* = 120) are showed as points in **f** and one line in **g**, respectively. Results are representative of three individual experiments.

**NSs by itself causes neurological disorders and animal death.** To evaluate the contribution of NSs to neuropathy associated with RVFV infection, adult BALB/c mice were inoculated with 10 and 100 pfu of RVFV or RVFV ΔNSs C13 by intracranial injection. With this protocol, we bypassed the need of NSs for viruses to reach the brain of infected animals. As expected, fluorescence microscopy after immunostaining of the newly synthesized viral nucleoprotein N showed that virus replication occurs in brain tissues regardless of the NSs presence (Fig. 10a, b). The presence of infectious particles in the brain indicated that viral replication led to the release of infectious progenies whether animals were infected by RVFV or RVFV ΔNSs C13 (Fig. 10c). Viral production however seemed to be less pronounced in the absence of NSs, probably due to the inability of the ΔNSs strain to counteract the innate immune response.

After intracranial inoculation of mice with a dose of 100 pfu, animals infected with RVFV all developed neurological disease. One mouse died and the others were euthanized due to paralysis, convulsions, or other signs of neuropathy on day 4 and 5 post-inoculation (Fig. 10d). Mice infected with RVFV ΔNSs C13 did not show any signs of neurological disorders over the experiment duration, though two mice were found dead on day 7. When a dose of 10 pfu was inoculated, results were similar; the kinetic was somewhat slower and the lethality slightly lower (Fig. 10e). After 10 days, most animals survived intracranial injection with RVFV ΔNSs C13 while only one subsisted after exposition to the virulent RVFV strain. From these data, we conclude that NSs is the major viral factor responsible for RVFV-associated neurological disorders and virulence.

## Discussion

Amyloidogenic proteins are prevalent in a large spectrum of hosts, ranging from bacteria to humans. Their aggregation into fibrils frequently relates to molecular and cellular dysfunctions, most of the time fatal in higher animals[3]. Amyloidogenic proteins are exclusively host-encoded and the only case of infection described for amyloid diseases was so far the acquisition through environmental exposure to prions[2].

In this study, we expanded the concept of amyloid fibrils to authentic viral infections. Our EM pictures showed that the typical thick nuclear filaments associated with RVFV infection are composed of many straight, nonbranched 12-nm-width NSs fibrillar assemblies. We established that NSs alone makes fibrillary aggregates and that these aggregates bind to the amyloid-marker dye ThS, whose interaction with amyloids depends on structures rich in β-sheets. We demonstrated in addition that NSs is strongly resistant to detergents, in line with a recent investigation that was unsuccessful to produce the full-length protein as it forms large aggregates in solution[28]. Our results are also in agreement with structural predictions suggesting that NSs has intrinsically

disordered regions and an amino-terminal domain with a high propensity to form β-strands[28,29]. Overall, we found that NSs assemblies meet the criteria of amyloids[30].

Our results revealed that NSs aggregates differed considerably in terms of organization and morphology depending on their subcellular location. They were, however, all composed of fibrils. Regardless of the cell line used, the aggregates appeared large in the nuclei and much smaller in the cytosol. It is not clear whether NSs, a small protein devoid of any classical nuclear localization signal, enters the nucleus through passive diffusion or host cell carriers[22,28]. The highly crowded environment in the nucleus results in local spatial constraints and volume exclusion effects that might promote NSs assembly into long, large fibril-bundles. Specific nucleus proteins also likely participate in this exaggerated fibrillization as NSs interacts with several nuclear factors in mammalian cells, e.g., SAP30 and TFIIH subunits[10]. The transport into the nucleus of infected cells may also prevent NSs from interactions with cytosolic proteins, such as specific chaperones and proteases, and in turn, favor the large filament formation.

Our video recordings and microscopy studies showed that the dynamics of NSs fibrilization was surprisingly fast as detectable in the nuclei of infected cells within 4–5 h only. The large NSs filaments grow within the nucleus while the globular-looking aggregates remained in the cytosol. In addition, nuclear bodies were found at the early stages of infection; they might represent alternative folds of NSs assemblies as those involved in the nucleation and proliferation of other amyloid fibrils[3]. Overall, these data indicate that the aggregation dynamics of NSs resembles that of the highly aggregation-prone amyloidogenic proteins, such as the mutant Huntingtin protein and other proteins containing expanded polyglutamine (polyQ) stretches[31].

The nuclear NSs fibrils were roughly parallel and straight at the beginning of infection. When NSs expression was maximal, i.e., at the latest stages of infection, individual fibrils were then difficult to depict and alternated from 8 to 22 nm in width. These structures were similar in aspect and dimensions to the twisted filaments reported for the amyloid fibrils formed by tau[16]. More structural investigations will be required to determine whether the electron-dense nuclear NSs structures correspond to a specific arrangement of the viral fibrils, or simply, result from the locally dense concentration of fibrils. A parallel between NSs and the double-helical ribbons found in the amyloid-β and tau fibrils however appears attractive[16,32]. In such a model, each individual NSs fibril would be composed of one or more protofibrils, with two distinct fibril global conformations because of their straight or twisted assembly.

Our SDD-AGE analysis of DTT-treated samples revealed that disulfide bonds were important to stabilize HMW NSs aggregates. The mutagenesis of specific cysteine residues in NSs selectively prevented the formation of fibrils and large fiber-bundles. This

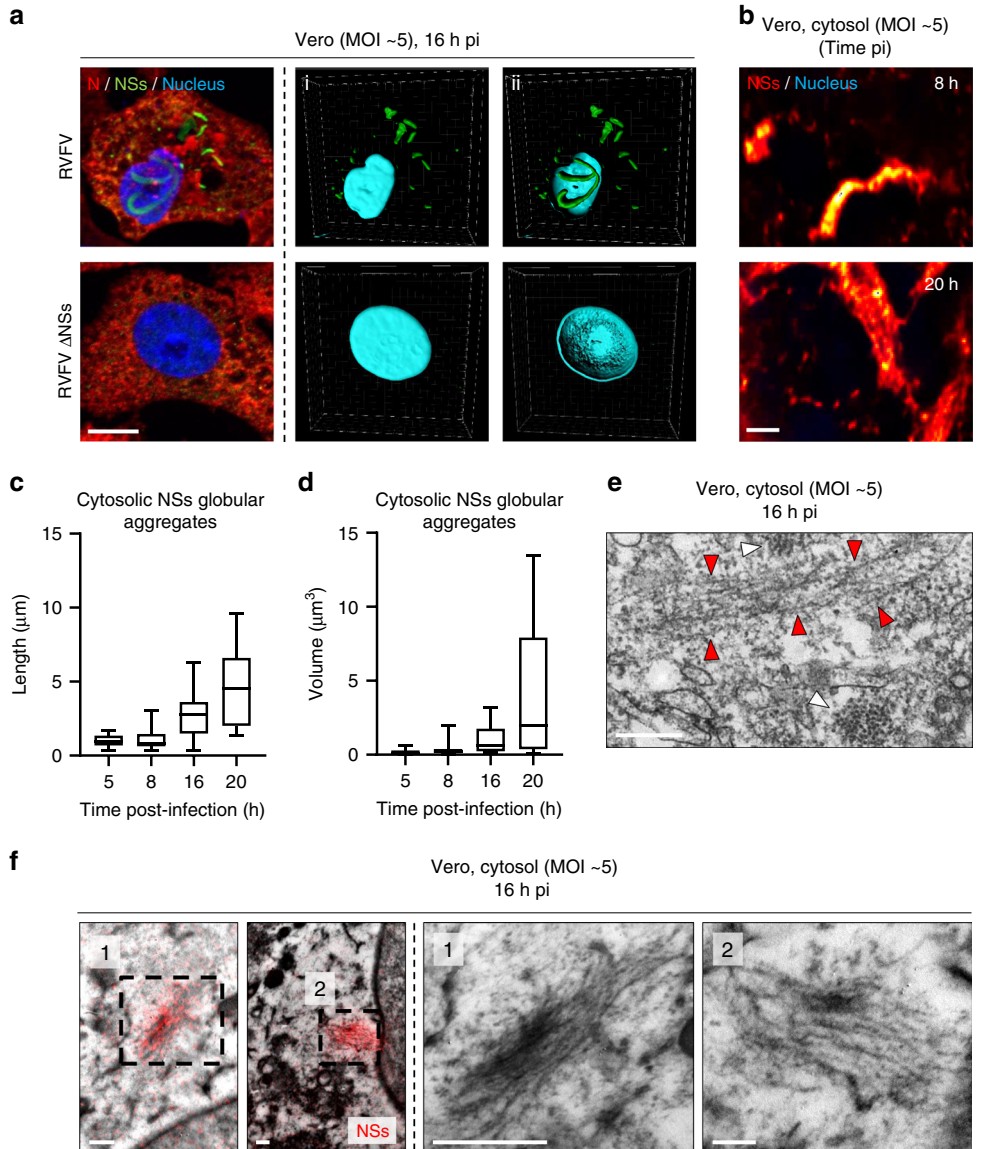

**Fig. 6 NSs assembles into short cytosolic amyloid-looking fibrillary aggregates. a** Vero cells were infected for 16 h with RVFV or RVFV ΔNSs, both at a MOI of 5. Infected cells were imaged by confocal microscopy after labeling of nuclei with Hoechst (blue) and immunofluorescence staining of intracellular RVFV proteins N (red) and NSs (green). The left panel shows Z-stack projections whereas the three others are 3D-reconstructions obtained with IMARIS software and displaying (i) cytosolic NSs and (ii) nuclear NSs. Results are representative of three independent experiments. Scale bar, 10 μm. **b** Vero cells were infected with RVFV for 8 and 20 h and subjected to immunofluorescence staining against NSs. Cytosolic NSs aggregates (red/yellow) and nuclei (blue) were imaged by STED microscopy. Images are representative of independent experiments. Scale bar, 1 μm. **c, d** Vero cells were exposed to RVFV (MOI ~5) for up to 20 h. The length (**c**) and volume (**d**) of cytosolic NSs globular aggregates was measured as described in Fig. 1h. $n = 13$, 41, 21, and 11 cytosolic NSs globular aggregates were examined at 5, 8, 16, and 20 h pi, respectively. Center line, median; box limits, upper and lower quartiles; whiskers, min. to max. range. **e** Vero cells were exposed to RVFV and imaged by TEM 16 h later. White and red arrowheads indicate cross- and longitudinal-sections of fibrils in the cytosol, respectively. Images are representative of three independent experiments. Scale bar, 200 nm. **f** Cytosolic aggregates imaged by CLEM following immunofluorescence staining against NSs (red). Fiber-arrays overlaid with the red fluorescence channel. Higher magnifications of cytosolic aggregates consisting of thin, short NSs fibrils are shown (black numbers and dashed squares). Experiments were repeated independently thrice with similar results. Scale bars, 250 nm.

further supports the view that redox reactions are important for NSs fibrilization. Although their exact ultrastructural fold remains to be solved, one could speculate that NSs fibrils are built from globular subunits such as β2-microglobulin and MLKL, which both form amyloid fibrils stabilized by disulfide bonds[19,33]. The oxidation of cysteine residues in NSs is likely stimulated by the viral factor itself. NSs in the mitochondria of infected cells leads to an early increase in reactive oxygen species (ROS)[34]. Increased ROS is known to compromise the redox control

systems in both the nucleus and cytosol, which in turn results in oxidative stress[35]. Furthermore, NSs provokes a S phase arrest in infected cells[36], a step in the cell cycle with the lowest reductive potential in the nucleus.

Of note, our results contrast with a previous report on the X-ray structure of NSs, for which a branched conformation was proposed with an all-helical fold and no intra- and intermolecular disulfide bond[28]. These authors used an avirulent strain of RVFV with several mutations in the NSs protein however, which might

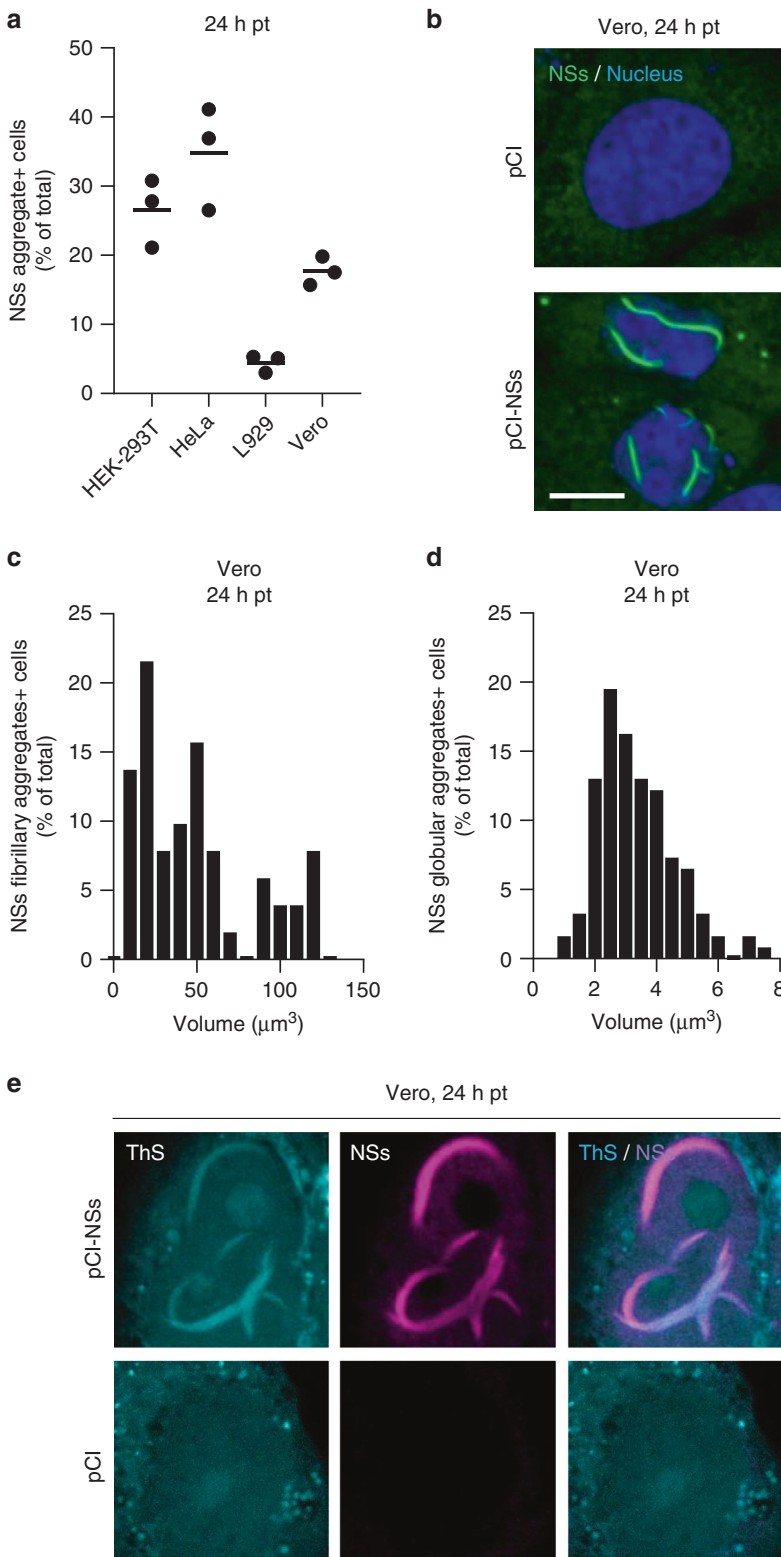

**Fig. 7 NSs expressed from plasmid DNAs forms nuclear and cytosolic aggregates. a** Various cell lines were transfected with the plasmid vector pCI encoding NSs. Cells were fixed 24 h post-transfection (pt), immuno-stained against NSs, and imaged by fluorescence microscopy. Images were analyzed for the percentage of total cells positive for NSs aggregates. Points represent independent experiments ($n = 3$). Center line, mean. **b** shows confocal images of transfected Vero cells with nuclei in blue and NSs in green. Images are representative of three independent experiments. Scale bar, 10 μm. **c**, **d** Transfected Vero cells were analyzed for the size distribution of NSs fibrillary (**c**, $n = 52$) and globular (**d**, $n = 164$) aggregates as described in Supplementary Fig. 2. **e** Transfected Vero cells were permeabilized with a Triton X-100-based buffer and then stained with the amyloid-binding dye ThS (blue) and Abs against NSs (purple) before confocal imaging. Experiments were repeated independently thrice with similar results. Scale bar, 5 μm.

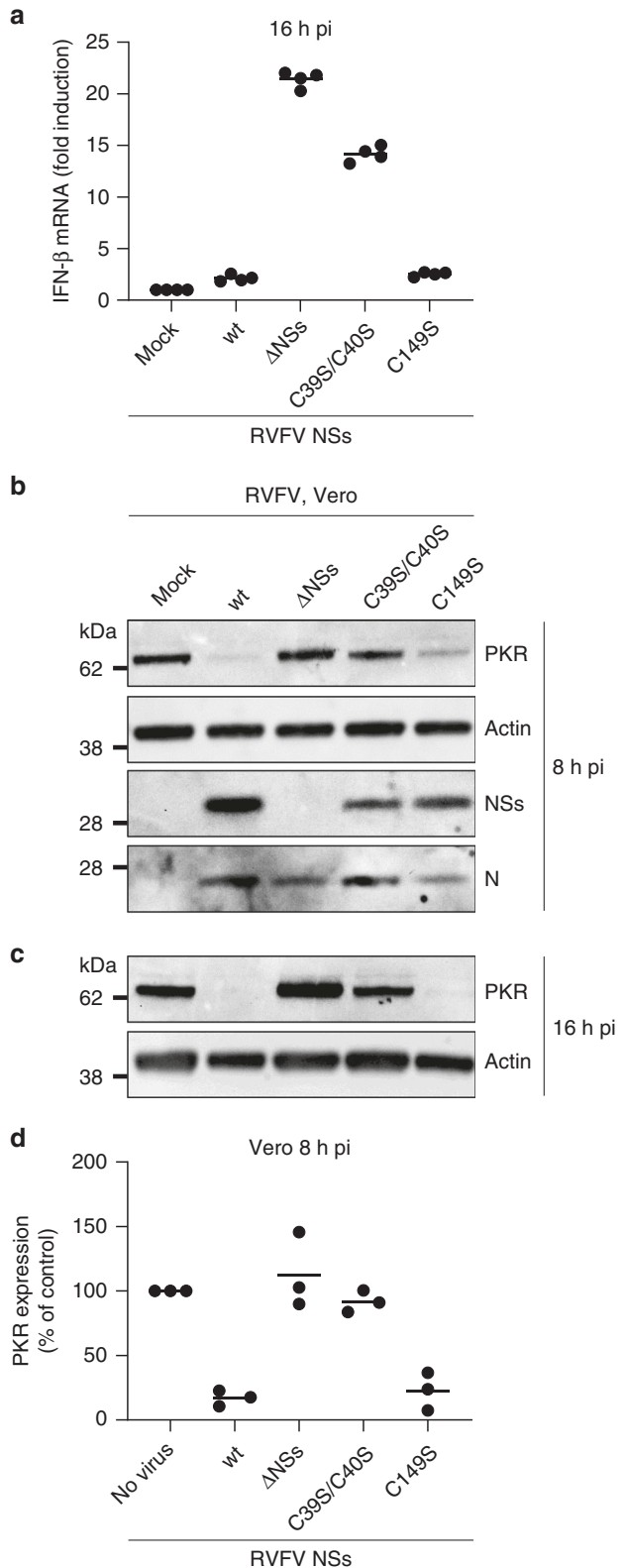

**Fig. 8 NSs fibrils suppress IFN responses. a** A549 cells were infected with either RVFV, RVFV ΔNSs, or the mutant viruses NSs C39S/C40S and C149S (MOI ~4) for 16 h. Infected cells were then lyzed and total RNA was extracted and purified. IFN-β mRNA levels were quantified by qRT-PCR. Points represent replicates (n = 4). Experiments were repeated independently twice with similar results. Center line, mean. **b–d** Vero cells were exposed to RVFV, RVFV ΔNSs, and the two mutant viruses NSs C39S/C40S and C149S (MOI ~5) for 8 h (**b**) and 16 h (**c**), and subsequently, lyzed and analyzed by SDS-PAGE and WB under reducing conditions with Abs against PKR, actin, NSs, and N. **d** Shows semi-quantification. Points represent independent experiments (n = 3). Center line, mean.

at the position 149 was important for the subsequent assembly into fiber-bundles.

Interestingly, RVFV appeared to rely on the NSs aggregation into short fibrils, but not into large nuclear fiber-bundles, for suppressing IFN responses. This is consistent with the capacity of NSs to counteract host cell defenses early after the beginning of infection[10]. The short NSs fibrils are undoubtedly formed even much earlier than fiber-bundles, which started to be visible after 4–5 h. Intriguingly, a recent study showed that mice survive infection by RVFV NSs C39S/C40S but succumb to exposition to RVFV NSs C149S[20]. Another group lately observed that mice perish from infection when RVFV encodes NSs molecules specifically mutated for lysine and threonine residues at position 150 and 152[37]. These mutations impaired the formation of nuclear NSs filaments but not of globular aggregates. Though the ultra-structure of these globular aggregates is missing, it is tempting to believe that they form short fibrils like the NSs mutant C149S. Altogether, the results support the possible link between NSs amyloid-like fibril formation, NSs biological functions, and RVFV virulence.

In our experiments, infected BALB/c mice developed the typical clinical signs of neuropathy when they were inoculated intraperitoneally with RVFV. We detected viral replication in the brainstem and found prominent nuclear NSs filaments in infected neural cells. In contrast, RVFV ΔNSs could not be detected in the brain and animals did not succumb to infection. When mice were injected by intracranial inoculation, RVFV ΔNSs was severely attenuated though productive viral replication occurred in the brain. Together these results highlight the critical role of NSs in neural spread and brain tropism in vivo, which in itself would be a reason for NSs being essential for the neuropathology. Our results further indicate that, besides functioning as a proviral and anti-IFN factor allowing the virus to replicate efficiently in vivo, NSs is even the direct cause of neuropathology, likely through the formation of presumably toxic amyloid-like fibrils. Further studies are nonetheless necessary to definitively ascertain the apparent correlation between amyloid formation and neurotoxicity.

Retinitis and hemorrhagic fever are other typical diseases occurring during RVFV infection[8]. The biological functions of thrombin and factor X, two clotting proteins, are impaired upon binding to amyloid fibrils[38], and both are believed to be potential targets of NSs[14]. Although the formation of NSs extracellular deposits in the retina and the release of NSs fibrils in the blood must be assessed in infected animals, it is tempting to postulate that NSs aggregation is also responsible for the virus-induced visual deficits and bleedings in infected patients. Moreover, NSs should be tested for its ability to seed the aggregation of host-encoded amyloid proteins.

NSs filaments are a signature of RVFV infection, but more than a hundred viruses are classified in the *Bunyavirales* order to

affect the folding or polymerization of NSs. Furthermore, this investigation was performed with a half-length truncated form of NSs, lacking the N- and C-terminal regions as well as the cysteines, which were most important for fibrilization in our study. The cysteine residues in the N-terminal part of NSs were indeed critical for the formation of short fibrils while the cysteine

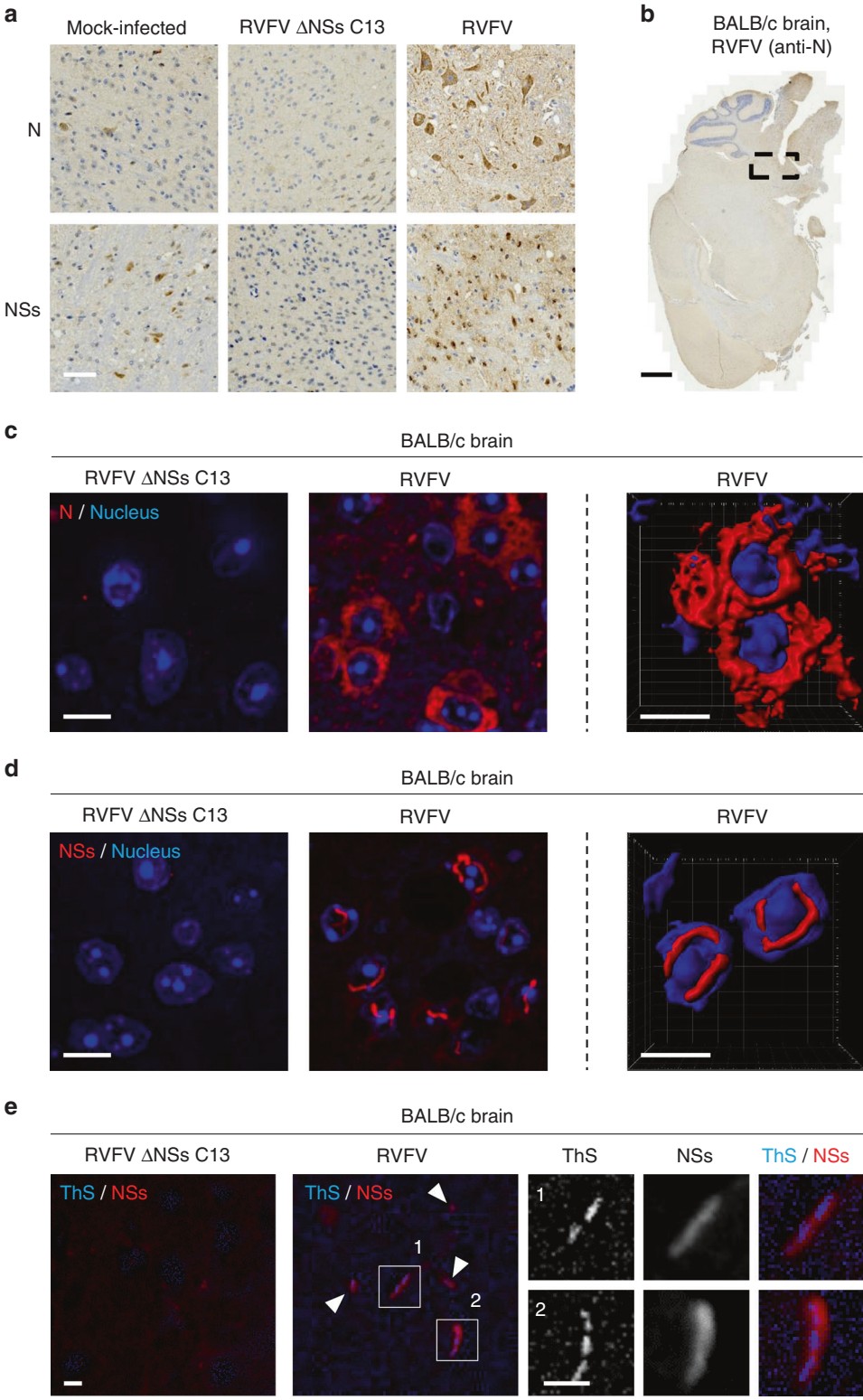

which RVFV belongs[39], and most code for an NSs-like protein[10]. The majority is poorly studied or not at all. Furthermore, polyoma- and adenoviruses have also been shown to encode proteins forming filamentous structures[40–42]. Although most of these viral proteins are still awaiting experimental characterization, it is likely that other viruses encode proteins able to form amyloid-like fibrils in vivo. The exact role of amyloid formation in the pathology of these viruses remains a challenge for future work.

## Methods

**Mice, cells, and viruses**. BALB/cByJ mice were purchased from Janvier Labs (Le Genest-Saint-Isle, France). All products used for cell culture were obtained from Thermo Fisher Scientific. The human and African green monkey kidney epithelial cells lines HeLa, HEK-293T, and Vero, as well as the murine L-929 fibroblastic cells and the human A549 lung and U-87 MG brain epithelial cells, were cultured according to ATCC recommendations. Baby hamster kidney cells stably expressing T7 RNA polymerase (BHK/T7-9 cells) were grown in minimal essential medium (MEM) supplemented with 10% tryptose phosphate broth, 5% fetal bovine serum (FBS), and 600 µg mL$^{-1}$ hygromycin. The RVFV strain ZH548 and its natural clone 13 (RVFV ΔNSs C13), which lacks most of the NSs sequence, were isolated

**Fig. 9 NSs forms nuclear filaments in the brain of infected animals. a** BALB/c mice were inoculated intraperitoneally with 100 pfu of either RVFV or its natural mutant clone 13 that lacks NSs expression (RVFV ΔNSs C13). When the first disease symptoms appeared, animals were sacrificed and brains collected, fixed, and subjected to immunohistochemistry staining against NSs or N. High magnification images of areas where viral replication occurs are shown. Results are representative of three independent experiments. Scale bar, 50 μm. **b** Shows a whole slide of a mouse brain infected with RVFV as analyzed in **a**. The black dashed box indicates the area of the brain where N and NSs were expressed. Note that only N staining is presented here. Scale bar, 1 mm. Brainstem tissues exposed to either RVFV or RVFV ΔNSs C13 were subjected to immunofluorescence staining against N **c** and NSs **d** and visualized by high-speed confocal microscopy (left panels). Series of Z-stacks were used to generate 3D-reconstructions with IMARIS software (right panels). The images show the cellular location of N **c** and NSs **d** in infected brainstem cells. Nuclei appear in blue and the proteins N **c** and NSs **d** in red. Experiments were repeated independently three times with similar results. Scale bars, 10 μm. **e** Brain tissues from mice infected with RVFV and RVFV ΔNSs C13 were permeabilized with a buffer containing Triton X-100 (0.5%) and Tween-20 (0.5%) before staining with ThS (1%) and Abs against NSs. Tissues were imaged with a fluorescence wide-field microscope. ThS appears in blue and NSs in red. White arrowheads show colocalizing ThS and NSs signals. Higher magnifications of NSs filaments are shown (white numbers and squares). Images are representative of three individual animals. Scale bars, 10 μm.

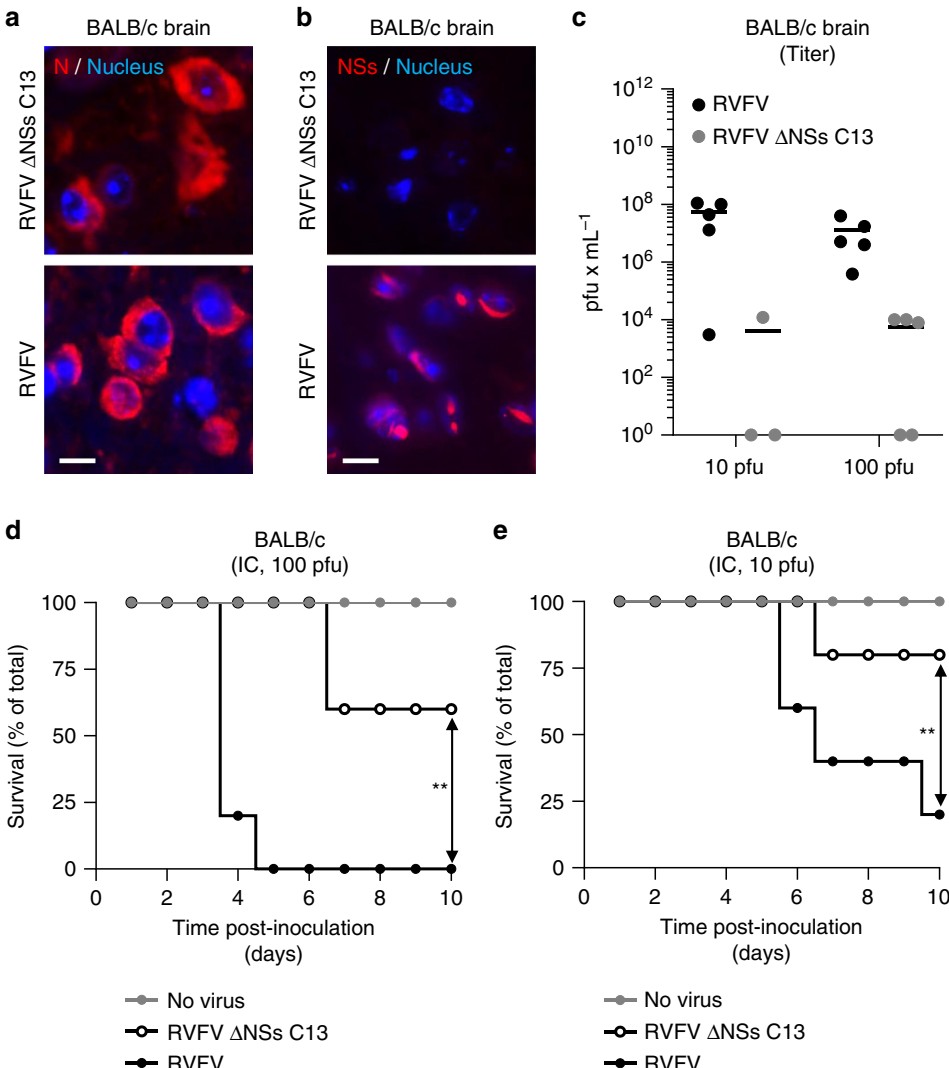

**Fig. 10 Expression of NSs in the brain is responsible for animal death. a–e** BALB/c mice were inoculated by intracranial injection with 10 and 100 pfu of RVFV and RVFV ΔNSs C13. **a, b** Brain tissues from animals infected with 100 pfu were stained with anti-N and anti-NSs Abs and visualized with a wide-field scanner. Nuclei appear in blue and the proteins N **a** and NSs **b** in red. Experiments were repeated independently thrice with similar results. Scale bars, 10 μm. **c** Brain from mice inoculated with 10 and 100 pfu of viruses were assessed for infectious particles by plaque-forming assay. Points represent titers of individual animals ($n = 5$ with the exception of the injection with 10 pfu of RVFV ΔNSs C13, $n = 3$). Center line, mean. pfu, plaque-forming units. Groups of five 9-week-old BALB/c male mice were inoculated with 100 **d** and 10 **e** pfu of viruses and survival of animals was monitored over 10 days. Simple linear regression tests were applied to assess the significance of differences observed between RVFV and RVFV ΔNSs C13. **\*\***$p = 0.0015$ **d** and 0.0029 **e**.

from human cases in Egypt and Central African Republic[43,44]. The recombinant RVFV lacking the full sequence encoding NSs (RVFV ΔNSs) was obtained by the genetic engineering of the RVFV ZH548 genome[45]. RVFV handling was achieved in a biosafety level-3 (BSL-3) lab. The virus stocks were obtained by harvesting the supernatant of Vero cells 72 h pi (MOI ~0.01). Titration was achieved by pfu assay. Briefly, following infection of confluent monolayers with ten-fold dilutions of virus, cells were grown in the presence of medium containing 2% FBS and supplemented with 0.9% agarose to abolish virus spread. Viral plaques were visualized and counted after staining with 0.2% crystal violet 5 days pi. The MOI is given according to the titer determined in Vero cells.

**Abs and reagents**. All Abs against RVFV proteins were made in the house[46,47] or kind gifts from N. Le May (IGBMC, France). Briefly, the mouse monoclonal Ab (mAb) 1D8 is raised against the RVFV nucleoprotein N. The rabbit polyclonal Abs (pAbs) SE2323 and 2284 are directed against the viral proteins N and NSs, respectively. The anti-PKR rabbit pAbs (18244-1-AP) was obtained from Proteintech. The mouse anti-α-Tubulin mAb B512 and anti-β-Actin mAb AC74 were both purchased from Sigma Aldrich. SDS was dissolved in water and ThS (Sigma Aldrich) in 50% ethanol.

**Plasmids, mutagenesis, and subcloning**. The plasmid coding for the RVFV protein NSs (pCI-NSs) was obtained by subcloning the DNA sequence encoding NSs between the unique NheI and KpnI sites in the polylinker of the plasmid pCI (Promega)[22]. A murine polymerase I (Pol I), five plasmids-based reverse genetics system was used to generate mutant and recombinant viruses[45]. Briefly, the system includes two expression plasmids, pTM1-N and pTM1-L that code for the RVFV nucleoprotein N and polymerase L, respectively, and three Pol I-driven plasmids, pRF108-S, pRF108-M, and pRF108-L that encode each the antigenomic RVFV RNAs (S, M, and L segments). The plasmid pRF108-S, which includes the virus S segment coding for the N and NSs proteins in an ambisense strategy[9], was used as a template for mutagenesis and to flank the tc peptide to the N-terminal region of NSs. To substitute cysteine by serine codons in the nucleotide sequence of NSs, site-directed mutagenesis was achieved by full amplification of the plasmid pRF108-S with the LA Taq polymerase (Takara) using the primer pairs C1-F/C1-R, C2-F/C2-R, C3-F/C3-R, and C4-F/C4-R and subsequent treatment with the methylation-dependent endonuclease DpnI (New England Biolabs). To generate a virus S segment encoding the tc-NSs, the sequence encoding the 3′ untranslated region, N protein, and the intergenic region (IR) was amplified by PCR with the LA Taq polymerase using the sense and antisense primers UTR-N-IR.F and UTR-N-IR.R. The sequence covering the IR, NSs, and 5′ untranslated regions was amplified by PCR, first with the primers IR-tc-NSs-UTR.F and IR-tc-NSs-UTR.R1 and then with the primers IR-tc-NSs-UTR.F and IR-tc-NSs-UTR.R2. The last antisense primer encodes the tc peptide (FLNCCPGCCMEP). Both 3′ and 5′ genomic amplified sequences were cloned into the TOPO plasmid (Thermo Fisher Scientific), sequenced, digested by BsmBI (New England Biolabs), and introduced into the pRF108 plasmid vector digested by BsmBI and BbsI (New England Biolabs). The complete list of primers used for mutagenesis and subcloning is shown in Supplementary Table 2.

**Cell infection**. Cells were exposed to viruses at indicated MOIs for 1 h at 37 °C. Virus input was then replaced by culture medium and cells incubated at 37 °C for up to 20 h. When infection was analyzed by light, TEM, and CLEM microscopy, cells were seeded, respectively, on coverslips, punched Aclar-Fluoropolymer films (EMS, Munich, Germany), and carbon-coated sapphire discs (Engineering Office M. Wohlwend GmbH, Sennwald, Switzerland), 1 day before infection.

**Cell transfection**. Vero cells ($5 \times 10^4$) were transfected with 0.8 μg of the plasmid pCI-NSs using Lipofectamine 2000 (Thermo Fisher Scientific) following manufacturer recommendations. Fresh culture media was added to the cells 6 h after transfection. Cells were fixed with 4% formaldehyde (FA) 24 h later.

**Confocal spinning-disc microscopy**. Infected and transfected cells were fixed with 4% FA, permeabilized with 0.1% Triton X-100 (Sigma Aldrich), and non-specific binding sites blocked with 5% bovine serum albumin (BSA). Incubation with primary Abs 1D8 and 2284 (both diluted 1:500) was followed by extensive washing and incubation with Alexa Fluor (AF) 488 or 568-conjugated secondary Abs (1:800, Thermo Fisher Scientific). Subsequently, cells were stained with Hoechst (1 μg mL$^{-1}$, Thermo Fisher Scientific) or 0.05% ThS for 120 min. When ThS was assessed, an AF647- instead an AF568-conjugated secondary Ab (1:800, Thermo Fisher Scientific) was used to detect the anti-NSs primary Ab 2284. Preparations were mounted in Mowiol (Sigma Aldrich) and data collected with a PerkinElmer VoX spinning-disc microscope equipped with a ×40 Nikon S Fluor oil immersion objective and the Volocity software v6.3 (PerkinElmer). Images were analyzed with ImageJ v1.52p (NIH, BSD license).

**Super-resolution STED microscopy**. Infected cells were subjected to immunofluorescence staining as the procedure described for spinning-disc imaging. STED microscopy was performed using a 2-color-STED microscope (Abberior

instruments GmbH) equipped with a ×100 Olympus UPlanSApo (NA 1.4) oil immersion objective. Nominal laser powers of 20 and 70% were applied for confocal excitation (594 nm) and STED (775 nm, 1.2 W), respectively. Pixel size was set to 60 and 15 nm for confocal and nondiffracted acquisitions, respectively. Minor contrast and brightness adjustments of images and Richardson–Lucy deconvolution (regularization parameter of $10^{-3}$, stopped after 30 iterations) were carried out using Imspector software 16.1.7098 (Abberior instruments).

**3D-reconstructions and analysis of confocal images**. 3D-reconstructions were obtained from the processing of spinning-disc and STED images, acquired as 0.3 and 0.2 μm Z-stacks respectively, by the IMARIS program v8.0.2 (Bitplane AG) and surface rendering and fluorescence thresholding tools. To define the number, type, size, and subcellular location of the various NSs assemblies, spinning-disc images were processed by the interactive deep learning and segmentation toolkit Ilastik 1.3.3 (University Heidelberg, GPL2 license)[15] as depicted in Supplementary Fig. 2.

**TEM**. Infected cells were embedded in epoxy resin for 50-nm ultrathin sectioning according to standard procedures. Briefly, cells were fixed in buffered aldehyde [2.5% FA, 2% glutaraldehyde (GA), 1 mM MgCl$_2$, 1 mM CaCl$_2$, 100 mM Cacodylate, pH ~7.2], post-fixed in 1% buffered osmium tetroxide, and en-block stained in 1% ethanolic uranyl acetate (UA). The adherent cells got flat-embedded in Epoxide (Glycidether, Serva). Ultrathin sections, contrast-stained with lead citrate and UA, were observed in a Zeiss EM 910 electron microscope (120 kV), and micrographs taken with image-plates, scanned at 30-μm resolution (Ditabis micron). Images were next analyzed with ImageJ.

**CLEM**. Cells grown and infected on discs were fixed with 4% paraformaldehyde and 0.2% GA in PHEM buffer (60 mM PIPES, 10 mM EGTA, 25 mM HEPES, 2 mM MgCl2, pH 6.9) before high pressure freezing, freeze substitution, and 200-nm-thin sectioning according to well-established protocols[48]. Saturation of unspecific binding sites with 0.1% fish skin gelatin and 0.8% BSA was followed by incubation with primary pAb anti-NSs (2284, 1:125) and then secondary AF568-conjugated Abs (1:100) and Hoechst (10 μg mL$^{-1}$). Grids were first imaged in deionized water using a Leica TCS SP8 confocal microscope with a ×63 HC PL APO (NA 1.4) immersion oil objective and the LAS X software (Leica), and then, with a Zeiss EM10 electron microscope (60 kV). CLEM analysis was achieved with the ec-CLEM plugin[49] of the Icy program v1.9.10.0 (Pasteur Institute, GPLv3 license)[50]. Single electron micrographs at 12,500-fold magnification were stitched in Image J using the "2D Stitching" plugin (standard settings, linear blending Fusion alpha 1.5)[51].

**Immunogold EM**. Sample preservation was achieved following the procedure used for CLEM analysis. Immunogold labeling was done by floating grids with 70-nm-thick sections on drops of blocking buffer (0.8% BSA and 0.1% fish skin gelatin in PBS) for 30 min. Grids were then incubated on drops of primary pAb anti-NSs (2284, 1:500 in blocking buffer) for 30 min and rinsed in PBS. Protein A conjugated to 10-nm gold beads (CMC university medical center Utrecht, Netherlands) was subsequently applied at a dilution of 1:50 for 20 min. The labeling was followed by extensive rinse, first in PBS, and then in water. Samples were finally post-stained using UA and lead citrate. The immunogold-labeled sections were imaged on a JEOL JEM-1400 electron microscope (JEOL, Tokyo) operating at 80 kV and equipped with a 4 K TemCam F416 (Tietz Video and Image Processing Systems GmBH, Gauting). Images were then analyzed by ImageJ.

**Nuclei purification**. Nuclei were isolated from sub-confluent Vero cells ($3.5 \times 10^6$) infected at a MOI of 5 for 16 h using the Nuclei EZ Prep kit (Sigma Aldrich) according to manufactures instructions. Purified nuclei were subjected to five cycles of sonication at high power for 60 s with a Bioruptor Plus sonicator (Diagenode), and then, cleared by 10,000-rpm centrifugation at 4 °C for 10 min.

**Protein analysis**. Infected cells were lyzed with a NP40-based buffer [25 mM TrisHCl (Sigma) pH ~7.5, 50 mM NaCl (Labochem International), 2 mM EDTA (Roth), 0.6% Nonidet P-40, 0.3% SDS, and 1× protease inhibitor (Roche)]. Total protein extracts were treated with DNAse I (Roche), and when specified, in addition subjected to 2%-lithium dodecyl sulfate (Thermo Fisher Scientific) treatment and heating up to 95 °C. Samples were then subjected to SDS-PAGE (Nu-PAGE Novex 10% Bis-Tris gel, Thermo Fisher Scientific) and transferred to polyvinylidenedifluoride membrane (iBlot transfer stacks, Thermo Fisher Scientific). Alternatively, SDD-AGE analysis was performed with a protocol derived from the work by Halfmann and Lindquist[18] as follow. SDD-AGE loading dye [5% glycerol (Honeywell Riedel-de-Haen), 0.05% bromophenol blue (Merck), and 1× protease inhibitor in 20 mM Tris (Roth), 10 mM acetic acid (Sigma), 1 mM EDTA, pH ~8.5 (0.5× TAE buffer)] and SDS at the indicated final concentration were added to cell lysates and nuclear extracts before incubation at up to 95 °C for 5 min. When indicated, proteins were reduced with 10 mM DTT (Sigma) for 30 min at 22 °C prior to SDS treatment. The samples were then loaded on an SDD-AGE [1.5% agarose (VWR Life Sciences) and 0.1% SDS in 1× TAE buffer], and the gel

run at 4 °C and 30 V for 18 h. Subsequently, blotting onto 0.2-μm nitrocellulose membranes (Roti®-NC, Carl Roth) by capillary transfer was performed using 20 mM Tris pH ~7.6 and 150 mM NaCl buffer at room temperature for at least 8 h. Subsequent incubation with primary pAbs 2284 or SE2323, respectively, against NSs and N (both 1:1000) or pAbs against PKR (1:1000) was followed by incubation with either anti-rabbit horseradish peroxidase- or alkaline phosphatase-conjugated secondary Abs (both 1:10,000, Jackson ImmunoResearch Laboratories and Vector Laboratories, respectively) and exposition to enhanced chemiluminescence reagents (SuperSignal from Thermo Fisher Scientific or ECF from GE Healthcare Life Science) according to manufactures recommendations.

**Rescue of the recombinant RVFV tc-NSs from plasmid DNAs**. The recombinant RVFV tc-NSs was rescued from the plasmid DNAs described above[45]. Briefly, viruses were recovered by transfecting BHK/T7-9 cells ($2.5 \times 10^5$) with the expression plasmids pTM1-L (0.5 μg) and pTM1-N (0.5 μg) together with 1 μg each of pRF108-S encoding tc-NSs, pRF108-M, and pRF108-L. Transfection was carried out with Lipofectamine 2000 (Thermo Fisher Scientific) using a ratio of 2 μL to 1 μg of plasmids in 1 mL of Opti-MEM (Thermo Fisher Scientific). Fresh culture MEM with 2% FBS was added to the cells 6 h after transfection. After 4 days, supernatants were collected, clarified, and titrated as described above.

**IFN-β mRNA quantification assay**. RNA was harvested from infected cells using NucleoSpin RNA extraction kit (Macherey-Nagel) as per manufactures instructions. cDNA was made using iSCRIPT reverse transcriptase (BioRad) from 250 ng of total RNA according to supplier instructions. Real-time quantitative reverse transcription PCR (qRT-PCR) was performed using primers IFN-β-mouse.F and IFN-β-mouse.R (Supplementary Table 2), SsoAdvanced SYBR green (BioRad), and following manufacturer recommendations. *HPRT1* was amplified with primers HPRT1-mouse.F and HPRT1-mouse.R (Supplementary Table 2) and used as normalizing gene.

**Live cell imaging**. Input virus was exchanged against phenol red-free medium with the ReAsH dye (200 nM) 3 h pi and cells imaged for 23 h in a biosafety level-3 environment. Data were collected with a Nikon Ti-Eclipse wide-field microscope equipped with a 20× Plan Apo lambda (NA 0.75) objective and the NIS Elements software v4 (Nikon) and analyzed with ImageJ.

**Mouse infection**. BALB/cByJ mice were maintained in BSL-3 isolators under 14:10 h day:night light cycle and unrestricted food and water. Nine-week-old males were injected intraperitoneally with 200 μL of culture medium virus-free for the control mice or containing 100 pfu of RVFV strains ZH548 or ΔNSs C13. Alternatively, animals were inoculated by intracranial injection with 10 μL of virus-free medium or medium containing up to 100 pfu of RVFV or RVFV ΔNSs C13. Mice were monitored at least daily for clinical symptoms and were euthanized as soon as the first clinical symptoms were noticed, due to the rapid evolution to death. Surviving mice were euthanized 10 or 14 days pi whether inoculated via intracranial or intraperitoneal injection, respectively. Animals were killed by cervical dislocation and brain dissected and fixed in 10% neutral-buffered formalin for 7 days.

**Imaging of mouse brain tissues**. After fixation, mice brains were embedded in paraffin for 3-μm sectioning. For histopathological analysis, staining with hematoxylin (Sigma Aldrich) and eosin (Leica Biosystems) (HE) was followed by incubation with primary rabbit Abs against N and NSs (1:100) and revealed with the Simple Stain MAX-PO kit (Histofine Biosciences inc., Cambridge, UK). For fluorescence wide-field microscopy, tissues were stained with Hoechst (1 μg mL−1, Thermo Fisher Scientific) and primary anti-NSs or anti-N rabbit pAbs (1:100) followed by extensive washing and incubation with secondary AF555-conjugated Abs (1:200, Thermo Fisher Scientific). Alternatively, brain sections were permeabilized with a buffer containing Triton X-100 (0.5%) and Tween-20 (0.5%) before staining with ThS (1%) and the primary rabbit pAbs against NSs on ice, in the dark, overnight. Slides were mounted in Fluoromount G mounting medium (Interchim) before being imaged with an Axio scan slide-scanner (Carl Zeiss Microscopy GmbH). Data were processed with the software Zen 2 (Carl Zeiss Microscopy GmbH) and analyzed with ImageJ. For confocal imaging, the brain slides were prepared as for fluorescence wide-field microscopy analysis, apart that secondary anti-rabbit AF568-conjugated Abs (1:800, Thermo Fisher Scientific) were used in combination with a Leica TCS SP8 confocal microscope with a ×63 HC PL APO (NA 1.4) immersion oil objective. Images were collected with the LAS X software and analyzed with ImageJ.

**Statistical analysis**. Graph plotting of numerical values, as well as the statistics, were achieved with Prism v8.4.1 (GraphPad Software). The sample sizes (n), reproducibility information, statistical methods, including parameters and p values, are indicated in the figure legends when appropriate.

**Ethics statement**. Mice experiments were conducted according to the French and European regulations on animal care and protection (EC Directive 2010/63/UE and French Law 2013-118 issued on 2013/2/1). The experimental protocol was approved by the Institut Pasteur Ethics Committee under #2016-0013 and authorized by the French Ministry of Research under #06463.

**Reporting summary**. Further information on research design is available in the Nature Research Reporting Summary linked to this article.

## Data availability
The material and data that support the findings of this study are available from the corresponding author upon reasonable request. The source data underlying Fig. 1g, h, 3b–f, 4a, b, d–i, 5c, d, f, g, 6c, 6d, 7a, c, d, 8a–d, and 10c–e and Supplementary Figs. 5C, D and 7A–C are provided as a Source Data file.

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

## Acknowledgements

This work was supported by grants from CellNetworks Research Group funds, Heidelberg, and from the Deutsche Forschungsgemeinschaft (DFG, LO-2338/1-1 and LO-2338/3-1). It was also supported by a Chinese Scholarship Council fellowship to Q.X. We thank H. Rezaei and J. Kartenbeck for support as well as N. Grandvaux, M. Moudjou, P. Shah, and N. Tischler for helpful discussions. We acknowledge V. Laketa and the Imaging Platform at the Center for Integrative Infectious Disease Research, Heidelberg. Immunogold labeling EM was performed by C. Funaya at the EM Core Facility of Heidelberg University. We would like to acknowledge her technical assistance. The ilastik team at the Heidelberg Collaboratory for Image Processing and D. Bucher are also gratefully acknowledged for their valuable help with computer-based image analysis.

## Author contributions

P.-Y.L. conceived the study, coordinated the research, and wrote the original draft of the paper; C.N.-K., H.-G.K., M.F., M.B., P.L., P.-Y.L., S.B. and X.M. designed the experiments; C.T., E.N., J.K., K.R., M.S., P.L., R.B., S.K. and X.M. performed the experiments; Q.X. carried out the computer-based modelings; All authors contributed to review and edit the paper.

## Competing interests

The authors declare no competing interests.
