## [Peer Review File · Nature Communications]

Reviewers' Comments:

Reviewer #1:

Remarks to the Author:

The authors present a detailed study demonstrating that the virus-induced filament-like aggregation of its virulence factor (the protein NSs) in cells exhibits characteristics typical for amyloids, which are known to lead to human brain diseases. This is an interesting and important result. The authors use a whole palette of biochemical and microscopy tools to convincingly show the proof. The manuscript is very well written and structured, and the experiments well thought and performed. I would really like to see this nice story published in Nature Communications.

I have a few comments though:

- Lines 70ff: The authors should make clearer from the start here that this is what has been shown in this manuscript (and not in previous work) – I got confused.
- General to STED and EM data: I would like to see comparing values on filament sizes also from the STED data at any time point and to compare it to the values from the EM data (should be resolvable in STED). In general, I would like to see more statistics on the values, maybe with specific plots (such as done in Figure 4, also add statistical significance values).
- Line 182: Are these the HMWs identified in the western blots? This was unclear to me at first.
- Line 213: Confocal or STED images? (Why only confocal (if the case)?)
- Lines 219ff: Are these experiments on fixed or living cells?
- Lines 229ff: How was fluorescence labelling done here? Live or fixed cells?
- Line 239: I did not understand the meaning of the last part of this sentence?
- Lines 245ff: Did the dye ReAsH produce any background that might introduce some additional biasing dynamics? I would like to see more statistics and values here instead of just one representative movie.
- Lines 266ff: Fixed and immunolabelled (how) cells?
- Lines 272ff: I would like to see an analysis/plots again like done for the previous measurements. How do the authors know that "these fibrils did not correspond to any of the known endogenous regular structures involving"?
- Line 293: I would like to see a statistical analysis with values for the aggregate sizes here as well.

Reviewer #2:

Remarks to the Author:

NSs is the main virulence factor of RVFV that, although being intensively researched, acts by several mechanisms that are not really understood. Léger et al convincingly showed by a series of methods that the NSs is the first example of a viral protein that forms amyloid fibers. A relevance for pathogenesis is not clear, but very likely.

Major points:

- There are several inhibitors of amyloid formation that would be interesting to try on NSs.
- From the effect of DTT on the resolution of NSs polymers it is concluded that NSs fibers are stabilized by disulfide bonds. However, the intracellular redox milieu is normally highly reducing. Upon cell lysis with buffers not containing reducing agents, oxidation happens quickly. So, strictly speaking, HMW smears on the SDS gel in the absence of DTT do not necessarily proof disulfide bond formation inside the cell. Of course virus infection means oxidative stress, so an increase in intracellular fibril growth may well be due to that. Experiments with antioxidants would help to solve this question. These results would also be important in the light of the discrepancies with the NSs structure paper by Barski et al., who did not find any disulfide bonds.

Minor points:

- The authors discuss the relevance of the fibrils for pathogenesis only in passing. Fibrils appear after 4 ½ hours, but many anti-IFN activities of NSs take place earlier than that. A more profound discussion of the literature is necessary.

Reviewer #3:

Remarks to the Author:

Leger et al studied the localization and structure of the NSs virulence factor of RVFV in vivo and in vitro. They unexpectedly found by TEM, various fluorescence-based microscopy including amyloid-specific dye and other techniques that NSs forms amyloid-like fibrillar structures in virally infected cell lines. Such amyloid fibrils are known to be disease-associated and may contribute to the virulence of RVFV. The fact that RVFV encodes an amyloidogenic protein is an important finding that offers interesting novel avenues for further research. However, there are some issues that dampen my overall enthusiasm of this fascinating finding.

1. What is the role of the virally-encoded fibrils? Authors speculate a bit in the discussion but this is vague. There are examples in the literature that e.g. herpesviruses encode proteins that form amyloids with cellular proteins to inhibit necroptosis (Pham et al 2019, Embo).
2. Authors state that RVFV is the first example of a virus that encodes amyloid. This is not correct (see paper above and McIntosh PB, J Virol. 2008 Aug;82(16):8196-203)
3. Evidence for the presence of amyloid fibrils in tissue section obtained from the RVFV infected mice is missing.

Minor issues:

What happens with the NSs fibrils after virus-induced cell death of RVFV-infected cells? I.e., are fibrils found in the medium of infected cell cultures (or even in the blood/liquor of infected animals?)?

Line 85 - What does "lacks most of the NSs" mean? Is something still being expressed? Is the virus hampered in infectivity?

Fig 1 - Why was ThS not stained in the histological sections? The direct evidence for amyloids in vivo is missing.

Line 91f/ Fig 1A/2A - As there is neither N or NSs visible in the staining of Δ NSs, is it possible that cells are not infected? In contrast to Fig. 1A, Fig. 2A shows N expression in the cytoplasm of U87 cells infected with the Δ NSs virus. Why not in Fig. 1A?

Line 105 - Is the formation of filamentous structures dependent on the MOI?

Line 126/ Fig 2 C-F - Why is there a switch from Vero cells (for STED) to HeLa cells (for TEM) back to Vero cells (for CLEM)? Why are different imaging methods used for each cell line tested?

Line 140 - Similar to earlier point - why is one cell imaged with STED and another with TEM? Why not Vero for both methods?

Line 143 - "None of the nuclear rosettes were found in cells infected with Δ NSs virus" - please show this

Fig S3A - Hard to distinguish between cell and fibrillar structure

Line 150/ Fig S3B - Why does this local annealing occur at such low frequencies?

Line 163 - Binding of ThS does not unambiguously indicate amyloid fibrils, as the dye is specific for cross β -structures

Fig 3A - Why are there globular structures visible in the ThS signal for cells infected with Δ NSs strain?

Line 186/ Fig 3C - Is the fraction of NSs in the wells significant compared to the amount of protein in each lane?

Line 201 - How do disulfide bridges form in the reductive environment of the nucleus?

Line233 - Fig 4 F and G Is there data available for in-between timepoints?

Line 291 - As Vero cells are poorly transfectable, the fact that only 5% of cells expressed NSs might also just be because only those were transfected, not a specific effect of NSs.

Peer reviewer 1 comments

The authors present a detailed study demonstrating that the virus-induced filament-like aggregation of its virulence factor (the protein NSs) in cells exhibits characteristics typical for amyloids, which are known to lead to human brain diseases. This is an interesting and important result. The authors use a whole palette of biochemical and microscopy tools to convincingly show the proof. The manuscript is very well written and structured, and the experiments well thought and performed. I would really like to see this nice story published in Nature Communications.

I have a few comments though:

Author response:

We first want to acknowledge Reviewer 1 for the enthusiastic and constructive comments about our work. This is highly appreciated.

Specific point 1. Lines 70ff: The authors should make clearer from the start here that this is what has been shown in this manuscript (and not in previous work) – I got confused.

Author response:

We have made clearer what is shown in our manuscript and rephrased the first sentence of the last paragraph of introduction (lines 76 – 79).

Specific point 2. General to STED and EM data: I would like to see comparing values on filament sizes also from the STED data at any time point and to compare it to the values from the EM data (should be resolvable in STED). In general, I would like to see more statistics on the values, maybe with specific plots (such as done in Figure 4, also add statistical significance values).

Author response:

Concerning the manuscript in general, we now show quantitative analysis of most microscopy-based approaches as well as the relevant statistical information (means, standard deviations, statistical tests, and p-values when applicable), namely Figures 1G, 1H, 3B, 3C, 3D, 3E, 3F, 5F, 5G, 6C, 6D, 7A, 7C, and 7D.

Regarding Specific Point 2, we now include a comparison of the values obtained for the filament width by EM and STED microscopy over an infection period of 16 hr (Figure 1G). We also show measurements of filament length by STED microscopy up to 20 hr post-infection (Figure 1H). In the latter case, we felt that it is not relevant to include measurements obtained by EM since the analysis was performed in 2D and did not provide information on the full length of filaments as these usually passed through many stacks. The text has been updated in the results accordingly and two paragraphs now describe Figures 1G and 1H (lines 123 – 128 and lines 138 – 148).

Specific point 3. Line 182: Are these the HMWs identified in the western blots? This was unclear to me at first.

Author response:

Yes, the HMW aggregates were those observed by SDS-PAGE. This point has been clarified in the text (lines 178 – 180).

Specific point 4. Line 213: Confocal or STED images? (Why only confocal (if the case)?)

Author response:

Images in Figure 4 were taken with a confocal microscope. This has been clarified in the text (lines 240 – 242). Confocal microscopy remains faster and easier for imaging many samples in 3D, which was necessary here to test multiple multiplicities of infection and time points. In addition, the size of NSs aggregates was so large that the limit resolution (about 200 nm) of confocal imaging did not really represent a limitation in this specific case. To complete this approach, STED microscopy and quantitative analysis of STED images are now shown all along the manuscript.

Specific point 5. Lines 219ff: Are these experiments on fixed or living cells?

Author response:

The analysis of these experiments was achieved with fixed samples. Only experiments involving the tetracycline tag and ReAsH dye were performed in living cells. This is now clarified in the text, lines 247 – 249.

Specific point 6. Lines 229ff: How was fluorescence labelling done here? Live or fixed cells?

Author response:

In this series of experiments, cells were fixed at different time points post-infection and immunostained with antibodies against NSs before confocal imaging and analysis. This point has been made clearer in the text (lines 256 – 257).

Specific point 7. Line 239: I did not understand the meaning of the last part of this sentence?

Author response:

The high density of NSs molecules in aggregates provides numerous binding sites for the ReAsH dye, all concentrated in a delimited volume and creating optimal conditions for the detection of tc-NSs aggregates. This point has been clarified in the text (lines 273 – 278).

Specific point 8. Lines 245ff: Did the dye ReAsH produce any background that might introduce some additional biasing dynamics? I would like to see more statistics and values here instead of just one representative movie.

Author response:

We now provide real-time quantitative analysis of tc-NSs aggregate formation in Figures 5F (number of NSs assemblies) and 5G (length of NSs filaments). Results with tc-NSs were consistent with those obtained in fixed cells after infection with the wt virus. We thus conclude that the tc peptide did not introduce noticeable bias in the dynamics of filament formation. Text has been updated to describe Figure 5F and 5G and state the absence of bias in the dynamics of filament formation (lines 279 – 283).

Specific point 9. Lines 266ff: Fixed and immunolabelled (how) cells?

Author response:

Cells were fixed and immunolabelled in these experiments. This point has been clarified in the text (lines 301 – 303).

Specific point 10. Lines 272ff: I would like to see an analysis/plots again like done for the previous measurements. How do the authors know that “these fibrils did not correspond to any of the known endogenous regular structures involving”?

Author response:

Cytosolic NSs globular aggregates have been imaged by STED microscopy after immunolabeling against NSs and examined for length and volume. Results are presented in Figures 6C and 6D. The text has been updated to describe the two new figures (lines 308 and 309).

We agree that our statement on NSs cytosolic fibrils was confusing. We meant that the NSs fibrils did not resemble any well-known cellular endogenous structures, in terms of shape and size. We clarified this point in the text (lines 313 – 315).

Specific point 11. Line 293: I would like to see a statistical analysis with values for the aggregate sizes here as well.

Author response:

NSs aggregates have been analyzed in transfected cells and we now show the volume distribution for both fibrillary and globular aggregates in Figures 7C and 7D. The text has been updated to describe the two new figures (lines 328 – 330).

Peer reviewer 2 comments

NSs is the main virulence factor of RVFV that, although being intensively researched, acts by several mechanisms that are not really understood. Léger et al convincingly showed by a series of methods that the NSs is the first example of a viral protein that forms amyloid fibers. A relevance for pathogenesis is not clear, but very likely.

Author response:

We thank the second Reviewer 2 for the pertinent comments that, we sincerely believe, helped to improve our investigation.

Major point 1. There are several inhibitors of amyloid formation that would be interesting to try on NSs.

Author response:

We agree with the reviewer that some molecules have indeed been proposed to prevent amyloid fibrilization. However, none of these molecules can target all amyloid proteins. Rather, each of them targets one particular amyloid protein and blocks fibrilization through very specific interactions that are unique to each disease proteins. Their capacity to cure protein misfolding diseases (PMDs) remains furthermore debatable and unfortunately, there is still no cure for any PMD.

The point raised by this reviewer is very important, and we intend to test multiple compounds for their ability to inhibit NSs amyloid aggregation and toxicity in future experiments. Such studies on the exact mode of action of an effective agent could help to elucidate how NSs amyloids are interfering with the host's immune defense. However, since this involves screening for many different compounds and concentrations, we feel that this type of analysis goes beyond the scope of the present study.

Major point 2. From the effect of DTT on the resolution of NSs polymers it is concluded that NSs fibers are stabilized by disulfide bonds. However, the intracellular redox milieu is normally highly reducing. Upon cell lysis with buffers not containing reducing agents, oxidation happens quickly. So, strictly speaking, HMW smears on the SDS gel in the absence of DTT do not necessarily proof disulfide bond formation inside the cell. Of course virus infection means oxidative stress, so an increase in intracellular fibril growth may well be due to that. Experiments with antioxidants would help to solve this question. These results would also be important in the light of the discrepancies with the NSs structure paper by Barski et al., who did not find any disulfide bonds.

Author response:

We thank Reviewer 2 for rising this important point. We addressed this point directly in cells and now provide a complete set of new data in Figures 3, S5, and S6. Briefly, DTT treatments and SDD-AGE analysis suggested that the large NSs fibrillar assemblies are substantially stabilized by disulfide bonds. To pursue this possibility, we replaced each of the cysteine codons by a serine codon in the NSs open reading frame and successfully rescued the four corresponding mutant viruses from plasmids, namely RVFV NSs C39S/C40S, C149S, C178S, and C194S (Figure S5). Cells exposed to RVFV NSs C39S/C40S and C149S did not have large filaments but rather amorphous aggregates (Figure 3). Our electron micrographs further indicated important ultrastructural differences between the aggregates made up of the two mutant proteins. While the NSs mutant C39S/C40S assembled into aggregates essentially built of small clumps, the NSs mutant C149S formed patches of short fibrils (Figures 3 and S6). These new results, together with our biochemical analysis, support a role of redox mechanisms in the formation of fibrils and the subsequent assembly into large nuclear filaments. Cysteines in the N-terminal region most likely drive NSs fibrilization while the one at position 149 promotes the subsequent growth of NSs fiber-bundles. The text has been updated in the results section (lines 199 – 230) and the discussion has extensively been rephrased to take these new findings into consideration.

Minor point 1. The authors discuss the relevance of the fibrils for pathogenesis only in passing. Fibrils appear after 4 ½ hours, but many anti-IFN activities of NSs take place earlier than that. A more profound discussion of the literature is necessary.

Author response:

We now included new data that (i) establish a link between NSs, amyloid-like fibrils (not fiber-bundles), and RVFV-induced suppression of IFN responses (Figure 8) and (ii) demonstrate a role of NSs in the neurological symptoms leading to death of animals (Figure 10).

(i) We have found that the NSs mutant C39S/C40S induces a strong IFN- β mRNA expression while the NSs mutant C149S is as efficient as the wt protein NSs to silence IFN- β mRNA (Figure 8A). Similarly, the C39S/C40S mutant was not able to induce the degradation of PKR while the C149S mutant of NSs was as efficient in reducing PKR levels as the wt NSs protein (Figures 8B to 8D). The NSs mutant C39S/C40S does not assemble into fibrils and filaments while the C149S mutant assembles into short fibrils but not into large filaments. From these results, we conclude that short fibrils (NSs mutant C149S), but not necessarily large fiber-bundles, are responsible for the anti-IFN activities of NSs.

(ii) We provide direct evidence for the role of NSs in neurological disorders during RVFV infection. To circumvent the limitation that RVFV clone 13 (a natural Δ NSs variant) cannot reach the brain, we inoculated mice by intracranial injection. The two viruses, RVFV and RVFV clone 13, replicated in brain tissues at similar levels (Figures 10A to 10B). Yet, mice infected with RVFV showed the typical neuropathy signs (paralysis, convulsions etc.), but not those inoculated with RVFV clone 13. The group of animals exposed to RVFV nearly all succumbed quickly, in contrast to the group of mice infected with RVFV clone 13 (Figures 10D and 10E).

We would like to highlight that NSs nuclear filaments (=fiber-bundles) are visible from 4-5 hr post-infection. From our results, it appears that short fibrils of NSs are a pre-requisite for the subsequent assembly into the large nuclear filaments. Therefore, the short fibrils are highly likely assembled within less than 4 hr, which is consistent with the capacity of NSs to block innate immunity early after the onset of cell infection.

The text describing this new series of data has been added in the results section (lines 343 – 364 and 402 – 422). The discussion has been extended in light of these new results, and now, also includes a more profound review of the literature.

Peer reviewer 3 comments

Leger et al studied the localization and structure of the NSs virulence factor of RVFV *in vivo* and *in vitro*. They unexpectedly found by TEM, various fluorescence-based microscopy including amyloid-specific dye and other techniques that NSs forms amyloid-like fibrillar structures in virally infected cell lines. Such amyloid fibrils are known to be disease-associated and may contribute to the virulence of RVFV. The fact that RVFV encodes an amyloidogenic protein is an important finding that offers interesting novel avenues for further research. However, there are some issues that dampen my overall enthusiasm of this fascinating finding.

We want to acknowledge Reviewer 3 for the valuable input that will improve the overall quality of our study.

Major point 1. What is the role of the virally-encoded fibrils? Authors speculate a bit in the discussion but this is vague. There are examples in the literature that e.g. herpesviruses encode proteins that form amyloids with cellular proteins to inhibit necroptosis (Pham et al 2019, Embo).

Author response:

Pham and colleagues show that a short peptide encompassing the RHIM domain of the herpesvirus M45 protein forms fibrils *in vitro* after being expressed in bacteria and purified in large quantities, which in our view is far from physiologically relevant. Though very interesting and important, the work by Pham et al. provides no evidence that the full-length M45 protein, or any other herpesvirus-encoded proteins, forms amyloid-like fibrils *in vivo* or in cell monolayers during the authentic viral infection.

We would also like to refer to our answer to the Minor Point 1 of Reviewer 2 on page 6 as this reviewer had also questions about the function of RVFV-encoded NSs fibrils and their role in virus-induced pathogenesis. Briefly, functions of NSs fibrils and their role in the pathogenesis are now further discussed under the light of our new findings, e.g. NSs short fibrils are required for counteracting IFN-responses and NSs in the brain of infected animals is responsible for neuropathy symptoms leading to death.

Major point 2. Authors state that RVFV is the first example of a virus that encodes amyloid. This is not correct (see paper above and McIntosh PB, J Virol. 2008 Aug;82(16):8196-203).

Author response:

We thank Reviewer 3 for referring to the two publications by Pham, MacIntosh, and colleagues. Both studies however show that truncated forms of genetically engineered viral proteins expressed from bacteria exhibit some amyloid characteristics *in vitro*. None of this work shows that the corresponding full-length wild type proteins form typical amyloid-like fibrils *ex vivo* and *in vivo*, i.e. during authentic viral infections in cell systems and animals. To the best of our knowledge, our study is the first comprehensive report on a wild-type protein encoded by a virus that spontaneously assembles into large amyloid-like fibrillary structures under conditions of an authentic viral infection, both in mammalian cells and in animals, **without any genetical engineering or chemical modifications**. We however agree with Reviewer 3 that these two references are important and deserved to be introduced and cited, which we have done now (lines 53 – 58). We have also changed the first sentence of the second and last paragraph in the discussion as indicated below to emphasize what is new in our manuscript (line 429 and lines 524 – 525, respectively).

Major point 3. Evidence for the presence of amyloid fibrils in tissue section obtained from the RVFV infected mice is missing.

Author response:

Our point was that NSs forms filamentous structures *in vivo* identical to those observed in cell monolayers, for which we provide strong evidence of their amyloid nature. It is therefore hard to believe that the properties of NSs are fundamentally different whether it is expressed in the brain of animals or in cell culture. We however understand the concern of Reviewer 3 and now include ThS staining of brain tissue sections. We would like to stress the important biosafety and technical limitations we had to face in performing such experiments. Amyloid staining *in vivo* is very challenging and working with RVFV represents an added complexity. We spent a significant time in establishing a method that provides convincing images and results. This is part of the reasons we had to ask for an extension, in addition to important restrictions due to the ongoing COVID19 pandemic.

RVFV is a BSL3 pathogen and classified among potential bioterrorism and MOT (which stands for “microorganism toxins”) in France, the country where the animal experiments in this study have been conducted. Due the local biosafety regulation, all infected organs have to be fixed in 10% formalin for 7 days before the samples can be taken out of the BSL3 animal house. These harsh conditions for fixation were not optimal to preserve the epitopes and molecular structures recognized by ThS. Next, the largest NSs aggregates are located in the nuclei of infected cells, and ThS poorly entered nuclei in tissue sections, consistent with our observations in cell monolayers. We assessed alternative amyloid-binding dyes, namely ThT, BTA-1, and NIAD4, but none gave better results. When harsher permeabilization protocols were used to make the dye accesses the nuclear content, the overall tissue integrity was lost.

We finally tested a protocol combining stronger, but not too harsh, permeabilization (Triton-X100 0.5% and Tween-20 0.5%) and higher concentrations of ThS (1%). Although a diffuse background noise was noticeable, we could observe that ThS only associates with NSs filaments in brain tissue sections from mice infected with RVFV. ThS did not bind to any other structures. Such colocalization events were absent in sections from animals exposed to the Δ NSs strain. The new results are shown in Figure 9E and the corresponding text has been updated in the results section (lines 393 – 401). Finally, we would like to emphasize that our new results, which show that only amyloid forms of NSs can affect the cellular interferon response, strongly suggest in addition that the formation of amyloid NSs fibrils in the brain is crucial to cause fatal neuropathy.

Minor point 1. What happens with the NSs fibrils after virus-induced cell death of RVFV-infected cells? I.e., are fibrils found in the medium of infected cell cultures (or even in the blood/liquor of infected animals)?

Author response:

One can imagine that aggregates are released in the outer medium or get degraded, even partially, as it has been described for some host-encoded amyloid fibrils (Chuang et al., J Cell Sci, 2018). Though these questions are particularly relevant to understand the virus-induced diseases, the first focus of our study is to demonstrate that NSs is an amyloid-like protein. It is a primordial prerequisite to move forward and address future questions, among others, on the details of molecular mechanisms leading to cell death and neuropathy symptoms. Many of these topics are under investigation in our lab and deserve dedicated, thorough investigations. To take into

consideration these nevertheless very interesting points of Reviewer 3, we have added a complete paragraph in the discussion (lines 516 – 523).

Minor point 2. Line 85 - What does “lacks most of the NSs” mean? Is something still being expressed? Is the virus hampered in infectivity?

Author response:

The natural mutant clone 13 (RVFV Δ NSs C13) has a large deletion in the NSs sequence and therefore NSs is not expressed at all. Similar to the genetically engineered Δ NSs virus, RVFV clone 13 is avirulent despite being not hampered in infectivity (Billecocq et al., 1996). The sentence has been rephrased and the original reference describing RVFV clone 13 has been added (lines 370 – 374).

Minor point 3. Fig 1 - Why was ThS not stained in the histological sections? The direct evidence for amyloids in vivo is missing.

Author response:

We kindly refer to our answer to the Major Point 3 of Reviewer 3 on page 9.

Minor point 4. Line 91f/ Fig 1A/2A - As there is neither N or NSs visible in the staining of Δ NSs, is it possible that cells are not infected? In contrast to Fig. 1A, Fig. 2A shows N expression in the cytoplasm of U87 cells infected with the Δ Nss virus. Why not in Fig. 1A?

Author response:

This point is very important and was not well enough highlighted in the original version of the manuscript. Brain cells were indeed not infected by RVFV Δ NSs clone 13, indicating that NSs is not only essential for the virulence but also for the neuronal tropism. This part has been rephrased (lines 379 – 386) and the fact that Δ NSs virus cannot reach the brain of infected animals is also addressed in the discussion (lines 507 to 515).

Minor point 5. Line 105 - Is the formation of filamentous structures dependent on the MOI?

Author response:

Yes, it is. Higher MOIs correlate with increased expression of NSs and increased size of the aggregates rather than a higher number. This is consistent with the proposed mechanism of amyloid growth by incorporation of monomers into the fibril. The results are shown in Figures 4, 5, S7, and S8 and Movie S1.

Minor point 6. Line 126/ Fig 2 C-F - Why is there a switch from Vero cells (for STED) to HeLa cells (for TEM) back to Vero cells (for CLEM)? Why are different imaging methods used for each cell line tested?

Author response:

NSs formed aggregates in all mammalian cell lines we tested (Figures 1A, 7A, and S1 and Table S1). Regardless of the cell line, the properties and dynamics of aggregate assembly were similar. Therefore, we equally used four cell lines representing human, monkey, and murine hosts (A549, HeLa, Vero, and L929, respectively) in our various imaging methods and other approaches. This strategy, which further demonstrates that NSs fibrilization is not cell type-specific, is now better explained in the text with two dedicated paragraphs constituting a specific chapter named “Formation of nuclear NSs filaments is supported by a wide range of cell lines” (lines 86 – 103).

Minor point 7. - Similar to earlier point – why is one cell imaged with STED and another with TEM? Why not Vero for both methods?

Author response:

Please see our answer to Minor point 6.

Minor point 8. Line 143 - “None of the nuclear rosettes were found in cells infected with Δ Nss virus” – please show this

Author response:

This negative control, i.e. HeLa cells infected with RVFV Δ NSs, is now shown in Figure S3C and described in the text (line 136).

Minor point 9. Fig S3A - Hard to distinguish between cell and fibrillar structure

Author response:

We replaced the original image with one with a better signal to noise ratio, which should make it easier to distinguish the NSs filaments from the remaining nuclear content.

Minor point 10. Line 150/Fig S3B - Why does this local annealing occur at such low frequencies?

Author response:

Local annealing increased over time and occurred with a high frequency 8 hr post-infection and later, correlating with increasing size of filaments. One can postulate that the local dense concentration of fibrils within fiber-bundles promotes a change in the overall fibril conformation, and in turn, the apparition of twisted-looking fibrils. However, more structural investigations will be required to determine whether the annealing corresponds to a specific form of the fibrils or simply results from locally dense concentration of fibrils. These points are emphasized in the results (lines 140 – 142) and in the discussion (lines 465 – 467).

Minor point 11. Line 163 - Binding of ThS does not unambiguously indicate amyloid fibrils, as the dye is specific for cross β -structures

Author response:

We fully agree with Reviewer 3, and we certainly did not attempt to claim that ThS binding provides an unambiguous indication of the amyloid nature of NSs. We believe that only a combination of independent, but complementary, approaches, involving ultrastructural microscopy methods and biochemistry techniques, can allow us to draw a conclusion about the

amyloid nature of NSs fibrillary and amorphous aggregates. This was exactly our strategy. The results (straight, nonbranched 12 nm-width fibrils, positive for ThS staining, strongly resistant to detergents etc.) are summarized in the second paragraph of the discussion (lines 429 – 439), and taken together, demonstrate that NSs meet all criteria of amyloids according to the International Society of Amyloidosis (Sipe et al., 2016, Amyloid).

Minor point 12. Fig 3A - Why are there globular structures visible in the ThS signal for cells infected with Δ NSs strain?

Author response:

Baer et al. (2011, JBC) showed that NSs provokes cell cycle arrest in phase S, and Benferhat et al. (2012, J Virol) reported that NSs filaments impact the overall organization of nucleus and chromatin. Our own electron micrographs indicate the exclusion of nucleoplasmic content from NSs filaments. From these results, it is evident that the nuclear organization dramatically differs whether cells express NSs or not. ThS signal in the nuclei of cells infected with the Δ NSs strain does not reflect amyloid staining, expectedly, but rather dye trapped in the nucleoli. The contrast with cells expressing NSs is magnified by (i) most nucleolar structures disappeared in those cells and (ii) the nucleolar stain got furthermore somehow dimmed because of the very strong ThS signal associated with NSs filaments. The molecular basis of ThS accumulation in intact nucleoli is difficult to explain and would deserve further investigations, but from our view, out of the scope of our study. We now comment on this point in the results (lines 162 to 167).

Minor point 13. Line 186/Fig 3C - Is the fraction of NSs in the wells significant compared to the amount of protein in each lane?

Author response:

We agree that the word ‘significant’ in this specific case is misleading since the insoluble fraction of NSs was essentially found in the gel, mainly as high molecular weight products. With ‘a significant fraction of NSs aggregates remained in the well’, we meant that part of NSs was still easily detected in the well. We have rephrased the corresponding sentence to clarify this point (lines 181 – 183).

Minor point 14. Line201 - How do disulfide bridges form in the reductive environment of the nucleus?

Author response:

There are examples of nuclear proteins that are oxidized in the nucleus, notably histone deacetylases (Matsushima et al., 2012, *Circulation Research*; Jansch et al., 2019, *Redox Biology*). Redox mechanisms in the nucleus are however quite complex and the molecular process leading to the oxidation of NSs would deserve a dedicated investigation. As mentioned by Reviewer 2, virus infection often means oxidative stress. NSs itself has been shown to be responsible for significant increase in reactive oxygen species (ROS) (Narayanan et al., 2014, *Virology*). ROS are well known to boost oxidative stress, also in the nucleus (Sies et al., 2017, *Annu. Rev. Biochem.*; Lukosz et al., 2010, *Antioxid Redox Signal*). Importantly, the level of reduction in the nuclear environment correlates with the cell cycle phase, the highest being observed during the G2/M phase, i.e. during cell proliferation (Lukosz et al., 2010, *Antioxid Redox Signal*). Interestingly, NSs was shown to cause cell cycle arrest in phase S (Baer et al., 2011, *JBC*). To follow the Minor Point 14 of Reviewer 3, and also the Major Point 2 of Reviewer 2, we now discuss the possible mechanisms that could promote NSs oxidation in the nucleus (lines 471 – 482).

Minor point 15. Line233 - Fig 4 F and G Is there data available for in-between timepoints?

Author response:

No in-between timepoints are available. However, this set of data is now completed with new results (Figures 5F and 5G) that show real-time quantitative analysis of the number and size of NSs filaments.

Minor point 16. Line 291 - As Vero cells are poorly transfectable, the fact that only 5% of cells expressed NSs might also just be because only those were transfected, not a specific effect of NSs.

Author response:

To answer this question, we have assessed four cell types (HeLa, Vero, L929, and HEK-293T) for NSs expression and aggregates formation following plasmid transfection. Next, we accurately quantified the number of cells positive for NSs expression and aggregates in each cell line (Figure

7A). We found that NSs could be detected in 5 to 30% of the cells, also in not more than 30% of HEK-293T cells, although transfection in this latter cell line is known to be very efficient. The text has been updated to include the description of these new results (line 332 – 335).

Reviewers' Comments:

Reviewer #1:

Remarks to the Author:

The authors have very well replied to my and in my opinion also the other reviewers' concerns and revised the manuscript accordingly. I have no further comments and suggest publication of this nice and important piece of work in Nature Communications.

Reviewer #2:

Remarks to the Author:

The points raised by this reviewer have been satisfactorily addressed.

Reviewer #3:

Remarks to the Author:

Authors made a great effort in addressing my concerns, by editing the manuscript, providing more background information on safety issues in work with virus, and they performed new experiments that strengthen the conclusion that NS forms amyloid to avoid innate immune response. I recommend to publish.

REVIEWERS' COMMENTS

Reviewer #1 (Remarks to the Author):

The authors have very well replied to my and in my opinion also the other reviewers' concerns and revised the manuscript accordingly. I have no further comments and suggest publication of this nice and important piece of work in Nature Communications.

Reviewer #2 (Remarks to the Author):

The points raised by this reviewer have been satisfactorily addressed.

Reviewer #3 (Remarks to the Author):

Authors made a great effort in addressing my concerns, by editing the manuscript, providing more background information on safety issues in work with virus, and they performed new experiments that strengthen the conclusion that NS forms amyloid to avoid innate immune response. I recommend to publish.

Author response:

We want to acknowledge all the reviewers for having taken the time to assess our revised manuscript and for their nice comments. This is highly appreciated. We sincerely believe that their valuable input helped to significantly improve our study in clarity and general interest.